# Ancient DNA reveals the lost domestication history of South American camelids in Northern Chile and across the Andes

Paloma Diaz-Maroto[1]*, Alba Rey-Iglesia[2], Isabel Cartajena[3], Lautaro Núñez[4], Michael V Westbury[2], Valeria Varas[5], Mauricio Moraga[6], Paula F Campos[7], Pablo Orozco-terWengel[8,9]*, Juan Carlos Marin[9,10]*, Anders J Hansen[11]

[1]The Saxo Institute, University of Copenhagen, Copenhagen, Denmark; [2]Section for Evolutionary Genomics, the GLOBE Institute, University of Copenhagen, Copenhagen, Denmark; [3]Faculty of Social Sciences, University of Chile, Santiago de Chile, Chile; [4]Institute of Archaeological Research and Museum, Católica del Norte University, San Pedro de Atacama, Chile; [5]School of Science Ecology and Evolution, Faculty of Sciences, Austral of Chile University, Valdivia, Chile; [6]Human Genetics Program, Institute of Biomedical Sciences, Faculty of Medicine, University of Chile, Santiago, Chile; [7]CIIMAR Interdisciplinary Centre of Marine and Environmental Research, University of Porto, Porto, Portugal; [8]School of Biosciences, Cardiff University, Cardiff, United Kingdom; [9]ICCMISAC - International Consortium for the Conservation Management and Improvement of South American Camelids, Cardiff, United Kingdom; [10]Genomic and Biodiversity Laboratory, Basic Sciences Department, Faculty of Sciences, Bio-Bio University, Chillán, Chile; [11]The GLOBE Institute, Faculty of Health and Medical Sciences, University of Copenhagen, Copenhagen, Denmark

*For correspondence:
palomafdm@gmail.com (PD-M);
Orozco-terWengelPA@cardiff.ac.uk (PO-W);
dromiciops@gmail.com (JCM)

Competing interests: The authors declare that no competing interests exist.

**Abstract** The study of South American camelids and their domestication is a highly debated topic in zooarchaeology. Identifying the domestic species (alpaca and llama) in archaeological sites based solely on morphological data is challenging due to their similarity with respect to their wild ancestors. Using genetic methods also presents challenges due to the hybridization history of the domestic species, which are thought to have extensively hybridized following the Spanish conquest of South America that resulted in camelids slaughtered en masse. In this study, we generated mitochondrial genomes for 61 ancient South American camelids dated between 3,500 and 2,400 years before the present (Early Formative period) from two archaeological sites in Northern Chile (Tulán-54 and Tulán-85), as well as 66 modern camelid mitogenomes and 815 modern mitochondrial control region sequences from across South America. In addition, we performed osteometric analyses to differentiate big and small body size camelids. A comparative analysis of these data suggests that a substantial proportion of the ancient vicuña genetic variation has been lost since the Early Formative period, as it is not present in modern specimens. Moreover, we propose a domestication hypothesis that includes an ancient guanaco population that no longer exists. Finally, we find evidence that interbreeding practices were widespread during the domestication process by the early camelid herders in the Atacama during the Early Formative period and predating the Spanish conquest.

## Introduction

The study of South American camelids (SACs) and their domestication has been a central subject in South American zooarchaeology since the 1970s. Many studies have focused on elucidating the domestication process as well as use and exploitation of camelid species in the past. These animals were extremely plastic, able to adapt to a great variety of environments found across the Andes and were even present in remote islands, emphasizing their importance in local ecosystems and for historic/prehistoric human populations. Camelids were key in the transition from hunter-gatherer to a mixed economy and played a central role in the cosmic vision of past Andean communities (*Wing, 1972*; *Stahl, 1988*; *Stanley et al., 1994*; *Wheeler, 1995*; *Olivera, 1997*; *Yacobaccio et al., 1998*; *Cartajena, 2003*; *Izeta et al., 2009*; *L' Heureux, 2010*; *Yacobaccio and Vilá, 2013*; *Gasco et al., 2014*).

There are currently four camelid species inhabiting South America: two wild species, guanaco (*Lama guanicoe*) and vicuña (*Vicugna vicugna*), and two domestic species, llama (*Lama glama*) and alpaca (*Vicugna pacos*). Guanaco and vicuña are the most important endemic large herbivores in South America due to their ecological dominance in the upper Andean ecosystem and their importance for human populations in terms of their subsistence strategies (*Muñoz and Mondini, 2008*). Llama and alpaca first appeared in the fossil record around 7,000 years before the present (yr BP) as the result of domestication carried out by human communities across the Andes (*Pires-Ferreira et al., 1976*; *Wheeler, 1984*; *Wheeler, 1995*).

Several hypotheses have been proposed for the origin of the domestic species (reviewed in *Wheeler, 1995*): (1) llamas were domesticated from guanacos and alpacas were domesticated from vicuñas, (2) llamas were domesticated from guanacos and alpacas derived from hybridization between llamas and vicuñas, (3) llamas and alpacas were both domesticated from guanacos while vicuña was never domesticated, and (4) llama and alpaca evolved from extinct wild precursors, while guanaco and vicuña were never domesticated.

To elucidate the origin and evolutionary history of camelids, numerous studies have performed DNA analyses on extant specimens (e.g. *Fan et al., 2020*; *González et al., 2019*; *Marin et al., 2013*; *Marín et al., 2017*; *Casey et al., 2018*; *Marín et al., 2007a*; *Wheeler et al., 2006*; *Kadwell et al., 2001*; *Stanley et al., 1994*). However, the conclusions drawn from these studies are ambiguous. Some mitochondrial DNA results have established the occurrence of hybridization in the past between wild and domestic species (*Wheeler et al., 2006*; *Stanley et al., 1994*; *Vidal-Rioja et al., 1994*). Others support the concomitant domestication of guanacos and vicuñas into llamas and alpacas (*Stanley et al., 1994*; *Kadwell et al., 2001*; *Wheeler et al., 2006*; *Marín et al., 2007a*). Finally, efforts using whole genome sequencing suggest support for the hypothesis that llama derived from guanaco and alpaca derived from vicuña (*Fan et al., 2020*).

Although the evolution and domestication of these species may be addressed using ancient DNA samples, such material is scarce for SACs. There are only few published studies based on ancient mitochondrial DNA (*Abbona et al., 2020*; *Díaz-Maroto et al., 2019*; *Gasco and Metcalf, 2017*; *Díaz-Lameiro, 2016*).

Osteometry has also been widely used in studies trying to understand the domestication of SACs. The first phalanx (anterior and posterior) and astragalus (*Yacobaccio, 2003*; *Gallardo and Yacobaccio, 2005*; *Izeta and Cortés, 2006*; *Cartajena et al., 2007*; *Cartajena, 2009*; *Izeta et al., 2009*; *L' Heureux, 2010*; *Reigadas, 2012*; *Reigadas, 2014*; *Gasco et al., 2014*) have been used for classification of two distinct size groups, the large sized guanaco and llama (genus *Lama*) and the small sized vicuña and alpaca (genus *Vicugna*) (*Wheeler, 1995*). However, due to the lack of significant differences between the domestic and wild camelids within each group (*Cartajena et al., 2007*), the taxonomic assignment based on osteometry within each genus remains challenging.

To evaluate the presence of domestic camelids in the Early Formative period (3,360–2,370 cal. yr BP) in Northern Chile and to elucidate the origin and evolutionary history of SACs, we have used a combination of osteometry and mitogenomics. We collected ancient samples in the archaeological sites of Tulán in Northern Chile and modern samples collected throughout the current distributional range of the vicuña and guanaco (*Figure 1*) following guidelines of the American Society of Mammalogists (*Sikes and Gannon, 2011*).

For the ancient DNA study, we have selected camelid bone samples from four archaeological sites (Tulán-54, Tulán-85, Tulán-52, and Tulán-94) from San Pedro de Atacama Desert, Northern

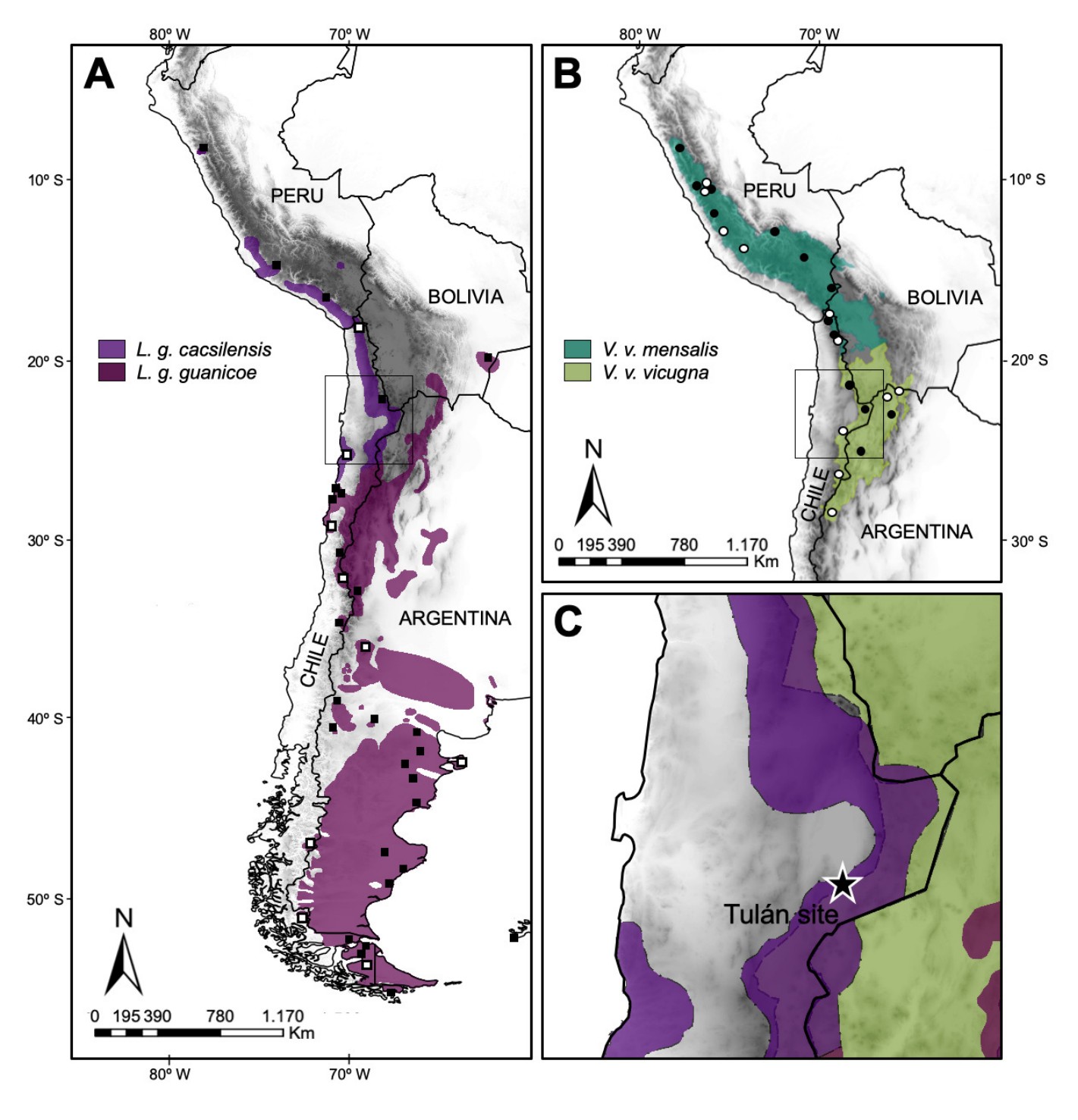

**Figure 1.** Map of the current geographic distribution for the wild species of South American camelids. (**A**) Map of the current geographic distribution of *Lama guanicoe* subspecies (based on *WCS, 2013*), white and black squares correspond to sampling locations of modern mitogenomes and control region (CR) hypervariable I domain, respectively. (**B**) Map of the current geographic distribution of *Vicugna vicugna* subspecies (based on *González et al., 2020*), white and black dots correspond to sampling locations of modern mitogenomes and CR hypervariable I domain, respectively. (**C**) Star shows the Tulán site in Atacama Desert in Chile, where the ancient samples (3,500–2,400 years before the present) were obtained.

The online version of this article includes the following figure supplement(s) for figure 1:

**Figure supplement 1.** Top: Study area and sites mentioned in the text (modified from *Núñez et al., 2009*).

**Figure supplement 2.** Main sunken ceremonial structure at Tulán-54 (left).

Chile. The two main sites, Tulán-54 and Tulán-85, belong to the Early Formative period where the camelid domestication is already incorporated into the economic activities by human groups of this area. Tulán-54 is located in Tulán ravine at the Atacama Desert and it shows the development of ceremonial and public architecture. Archaeological remains from offerings and domestic activities evidence that Tulán-54 was an important place of cultural and economic transformation from hunter-gatherers to early pastoralist communities in the highlands of the Southern Andes during the Early Formative society (ca. 3500–2400 BP) (*Núñez et al., 2017b*). Tulán-85 is an Early Formative site in the Tulán ravine, located on the border of the resource-rich salt flat. This site has been interpreted as a domestic area with few residential structures (*López et al., 2013*).

The comparison between the patterns of genetic variation of this dataset of ancient (61) and modern (66) mitogenomes along with the largest dataset of SAC mitochondrial D-Loop sequences (815) generated thus far provides a unique and deeper insight into one of the biggest unsolved mysteries of South American zooarchaeology, SAC domestication.

## Results

### Morphometric analysis of the phalanx and astragalus

Morphometric analysis of the astragalus and anterior and posterior phalanx shows two differentiated groups based on body size, small sized and big sized individuals, consistent with observations in the wild *Lama guanicoe* and *Vicugna vicugna* (*Wing, 1972*; *Kent, 1986*; *Elkin et al., 1991*; *Cartajena, 2003*; *Cartajena et al., 2007*). The small sized cluster most likely represents alpaca and vicuña specimens, while guanaco and llama individuals most likely form the big sized cluster. Archaeological measurements are consistent with modern camelids measurements (shown in *Figure 2*; *Cartajena, 2003*; *L' Heureux, 2010*). Nevertheless, problems on taxonomic assignation through osteometric techniques persist due to the lack of significant size differences between domestic and wild animals (*Lama glama/Lama guanico*e and between *Vicugna pacos/Vicugna vicugna*) and the use of models based on modern species dimensions, ignoring the large period of selection and change that cannot be recognized through current standards (*Moore, 1989*). Finally, there are wide intersection areas due to intraspecific variability; in some cases, subspecies have been defined with important size differences, which get confused interspecifically (*Novoa, 1984*; *Elkin et al., 1991*; *Cartajena et al., 2007*). By integrating modern species measurements into the scatterplots, size variability among the individuals stands out, depending on the origin of the animals (Table S4).

A statistical comparison of the two groups (i.e. large vs. small sized animals) for either of the bones measured (i.e. first anterior phalanx, astragalus) showed significant differences between the two groups of camelids (all p-values smaller than 10E-4). These comparisons were carried out between the two groups for each of the four morphological measurements (i.e. breadth distal [Bd] and greatest length medial [GLm] for the astragali, and breadth of proximal articulation [BFp] and depth proximal [Dp] for the first anterior phalanx), as well as on the product measurement of the two measurements for each bone (i.e. Bd (mm) * GLm (mm) for the astragali, and BFp (mm) * Dp (mm) for small specimens).

### Ancient mitogenome sequencing

Using a capture strategy to enrich our aDNA libraries for mitochondrial sequences, we generated 61 near complete mitogenome sequences from camelid bone samples collected in the Tulán ravine and Atacama Salar: 47 from Tulán-54, 12 from Tulán-85, 1 from Tulán-94, and 1 from Tulán-52. The average depth of coverage in the assembled mitogenomes ranged between 2.53x and 153.03x with a final length of 15,438 base pair (bp).

The analysis with mapDamage showed the characteristic pattern of ancient DNA consistent with postmortem deamination damage in all aDNA samples (*Figure 3—figure supplement 1*). An increasing presence of C-T and G-A mutations at the terminal ends of sequenced molecules was detected at the 5´-ends and 3´termini, respectively, as expected from the damage model used, generating an excess of cytosine deaminations at single-strands ends of the DNA templates.

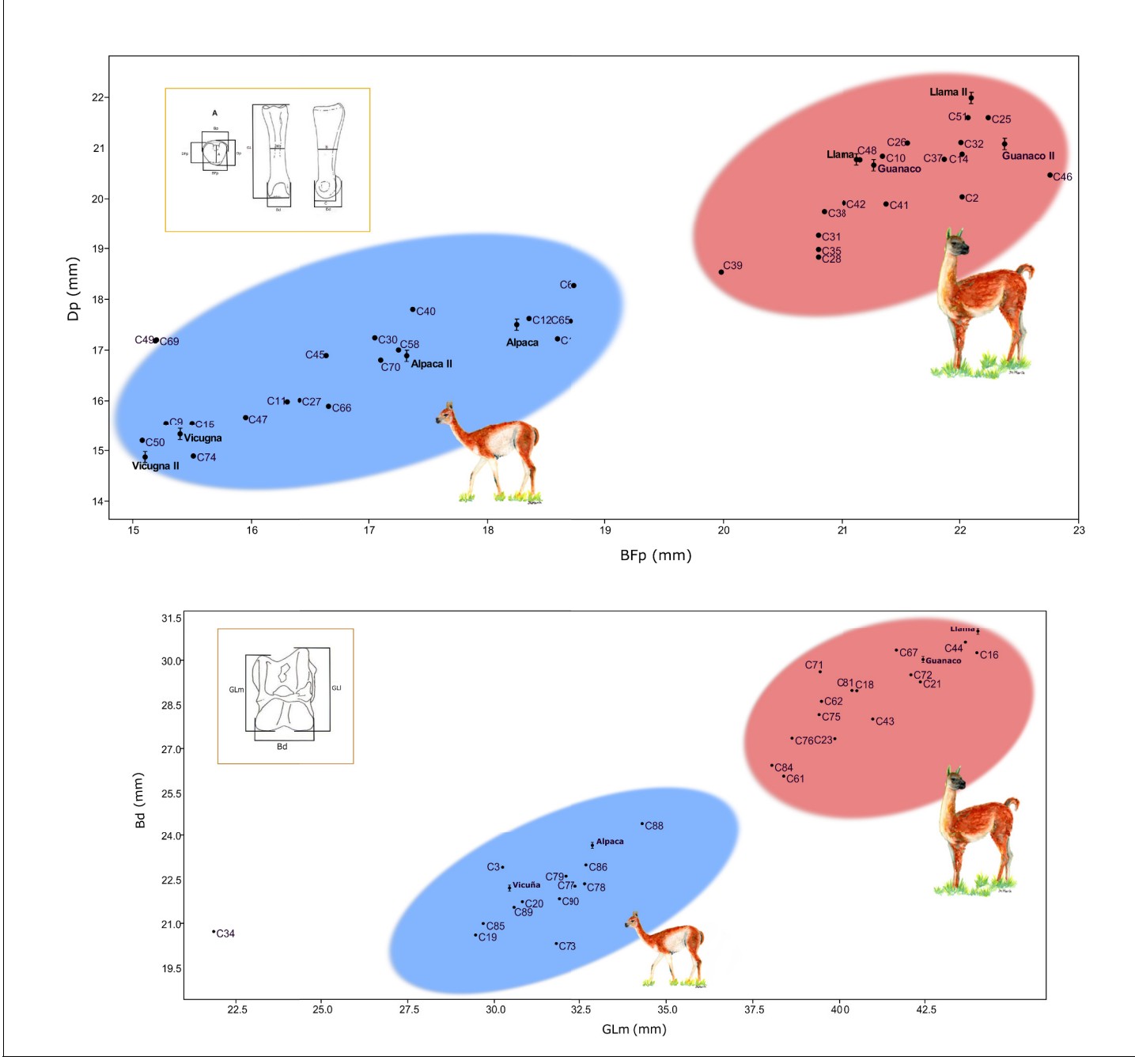

**Figure 2.** Correlation graphics of the first anterior phalanx and astragalus measurements. (a) Correlation for the anterior phalanx measurements; Dp = depth proximal and Bfp = breath of facies articularis proximalis. (b) Correlation for the astragalus measurements; Bd = breadth distal and GLm = greatest length medial. Modern camelids are shown in bold font with standard deviations estimated for the anterior phalanx and astragalus (*L' Heureux, 2010*; *Cartajena, 2003*). Modern samples measured by *Cartajena, 2003* were named as Guanaco II, Llama II, Vicuña II, and Alpaca II on the phalanx graphic. The blue ellipsis indicates the group of small sized specimens and the red ellipsis indicates the group of large sized specimens.

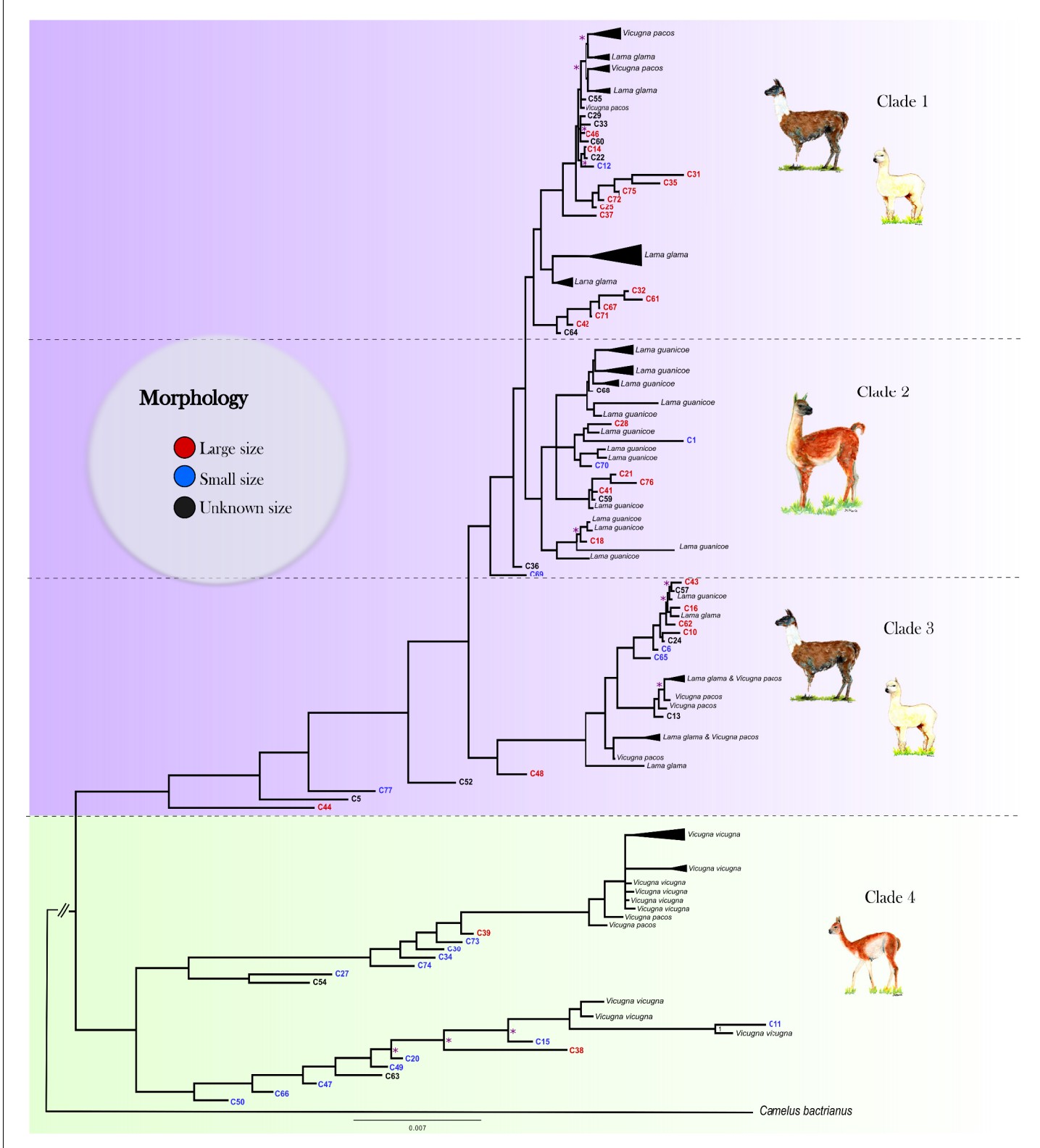

**Figure 3.** Mitogenomic phylogenetic tree. Bayesian reconstruction of phylogenetic relationships between ancient and modern camelids using complete mitochondrial genome data. For the aDNA samples, their identification code appears on the branch tip with a color indicative of their size class assignment based on the sample's morphological analysis. For the modern samples, the species' name and sample ID are in black font. Posterior probability values for 88.76% of the nodes were higher than 0.9, indicating strong statistical support for those nodes. The nodes with less than 0.9

*Figure 3 continued on next page*

*Figure 3 continued*

posterior probability are indicated with a pink asterisk on the tree. See *Figure 3—figure supplement 2* for a complete view of the modern samples included in the analysis.

The online version of this article includes the following figure supplement(s) for figure 3:

**Figure supplement 1.** DNA fragmentation and nucleotide mis-incorporation plots for samples from Tulán-54, Tulán-85, Tulán-52, and Tulán-94.
**Figure supplement 2.** Mitogenomic phylogenetic tree.
**Figure supplement 3.** Dated mitogenomic phylogenetic tree.

## Phylogenetic analysis

The Bayesian phylogenetic analysis of the ancient and modern mitogenomes using the Bactrian camel as outgroup resulted in strong statistical support for four clades in our data (*Figure 3*, *Figure 3—figure supplement 1*). Clades 1, 2, and 3 form a large monophyletic clade supported by a posterior probability of one with all modern domestic camelids located in Clades 1 and 3. Clade 1 consists of a mixture of modern llama and alpaca mitogenomes and 20 ancient samples (3 from Tulán-85, 16 from Tulán-54, and 1 from Tulán-52) with 13 individuals morphologically identified as large sized animals and 1 specimen as small sized. Most modern guanacos grouped within Clade 2, together with five samples from Tulán-54 morphologically identified as big sized and three samples as small sized animals. Clade 3 consists of both domestic camelid species and includes one modern guanaco, as well as five ancient samples (C44, C5, C77, C52, and C48) that are closer to the ancestral mitogenome of the big clade. Clade 4 was supported by a posterior probability of one and grouped the majority of modern wild *V. vicugna* and two alpacas (*V. pacos*) (A_1557_alpaca and A03_alpaca). In this clade, 12 ancient specimens (11 from Tulán-54 and 1 from Tulán-85) were morphologically identified as small sized camelids and two samples as big sized (C38 and C39 from Tulán-85). The haplotype diversity for the ancient samples was 1.00 and for the modern samples, the values oscillated within 0.971–0.993 (Table S8).

The dated BEAST phylogeny recovered the same major clades as the MrBayes analysis, lending support to our overall tree topology (*Figure 3—figure supplement 3*). In addition to the topology, we were also able to add approximate divergence dates of these main clades as well as the nodes within the clades. Clade 4 (vicugnas) diverged first from the rest of the SACs at ~1.01–1.18 Myr. Following the first divergence, Clade 3 forms at ~159–201 kyr with the final split forming Clades 1 and 2 at ~164–198 kyr. Within the clades, we uncover a number of different divergence dates between putative domestic and wild mitochondrial haplotypes.

The Bayesian Skyline Plot (BSP) obtained from modern and ancient maternal sequences shows an increase in the Ne of the domesticated clades (Clades 1 and 3) around 8,000–6,000 yr BP in accordance with the putative initiation of domestication. Clade 2 (guanacos) and Clade 4 (vicuñas) also present evidence of a demographic expansion, however at a much earlier time ~ 50,000 yr BP (*Figure 4*). Nevertheless, these two clades also show a modern demographic expansion, albeit less pronounced than for Clades 1 and 3, around the domestication time.

## Control region haplotype network analysis

To investigate the relationship between the ancient and modern camelids, we analyzed 300 bp of the CR hypervariable I domain. Among the 849 samples analyzed, we detected 76 different polymorphic sites and 158 haplotypes divided into two main haplogroups (*Figure 5*). Thirteen haplotypes were exclusively found in ancient samples, and eleven clustering with the *Lama guanicoe* and domestic camelid haplogroups: *cacsilensis* (66), *cacsilensis-guanicoe* (30), *guanicoe* (76), llamas (25), and alpacas (5) (Table S7).

## Temporal network suggests maternal lineage replacement in modern alpacas

A temporal network analysis of the 300 bp of the CR hypervariable I domain was performed using the same dataset as in the PopArt analysis. In total, 849 sequences were included, yielding 158

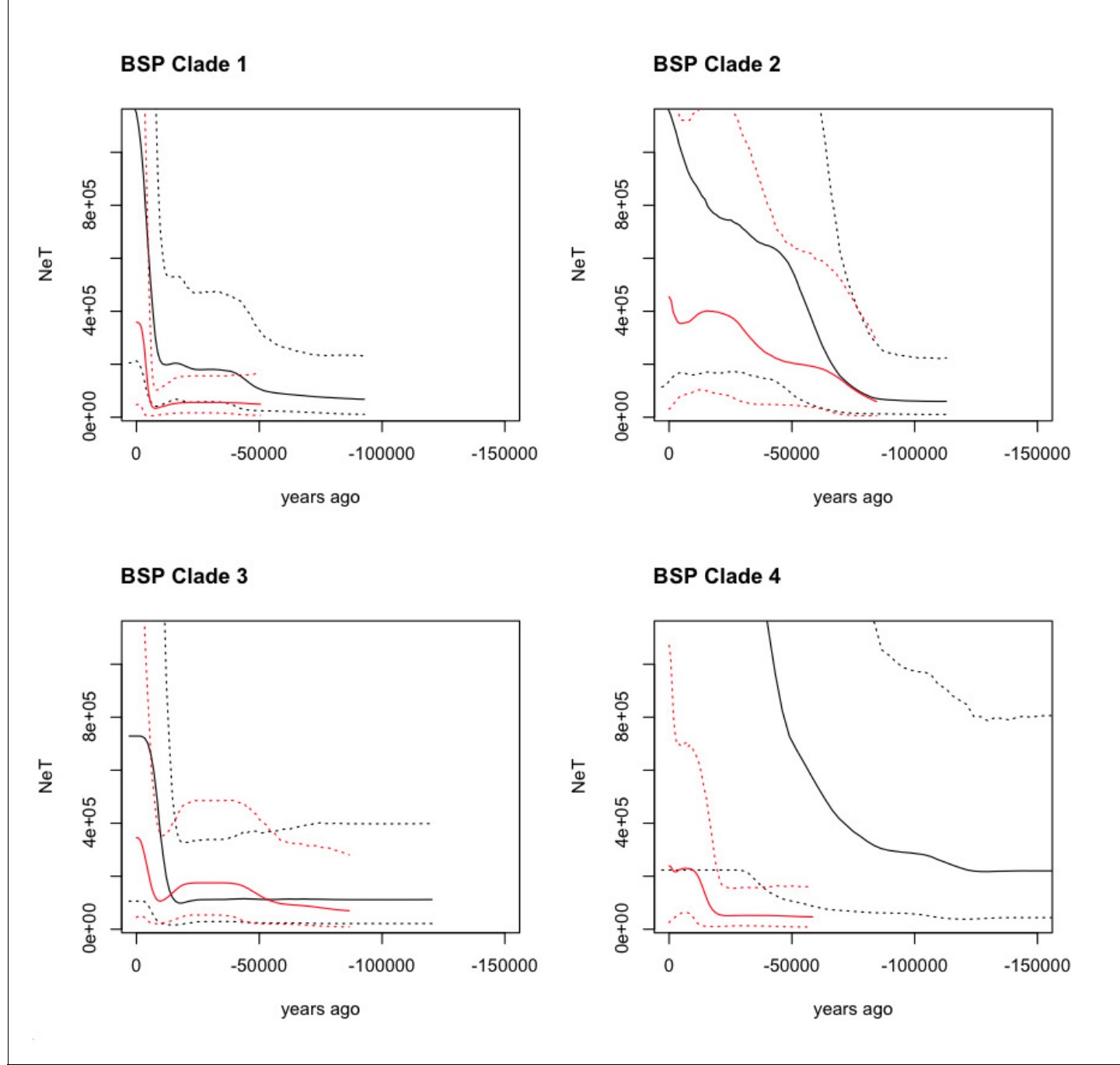

**Figure 4.** Bayesian Skyline Plot (BSP) derived from the analysis of the ancient and modern camelid data. The x-axis is units of years in the past (from the present - zero, toward [-] 150,000 years ago), and the y-axis is equal to NeT (the product of the effective population size and the generation length). The black line is the ancient DNA dataset and the red line is the modern DNA dataset with the dotted lines indicating the 95% confidence interval of the model and the solid line the mean.

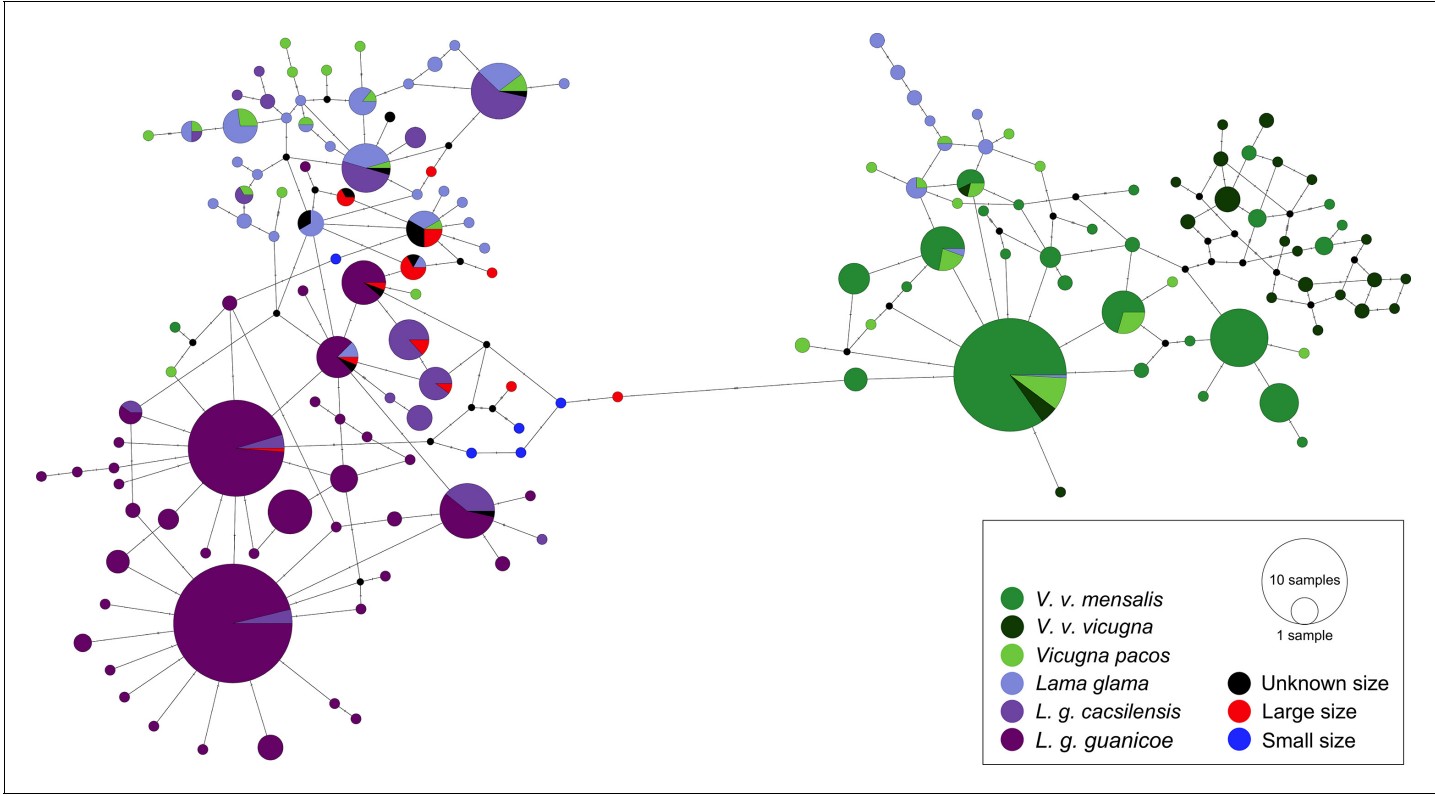

**Figure 5.** Minimum spanning network of ancient and modern camelid partial control region sequences. A total of 158 different hypervariable I domain haplotypes found among 849 sequences are shown. Each color represents a camelid subspecies and the ancient sequences. The haplotypes colored in dark purple include the two subspecies of *Lama guanicoe* (*cacsilensis* and *guanicoe*) and light purple represents *Lama glama*. The groups colored with dark green are formed by the two subspecies of *Vicugna vicugna* (*vicugna* and *mensalis*), while the light green includes the domestic alpacas, *Vicugna pacos*. The aDNA samples from this study are shown in black, red, and blue. Each haplotype is represented by a circle with its size proportional to the haplotype's frequency. Mutations are shown as small perpendicular lines crossing edges connecting haplotypes.

different haplotypes (42 from ancient sequences and 150 from modern sequences) of which only 11 haplotypes were shared by ancient and modern SAC despite the inclusion of a much higher number of modern mitochondrial sequences than ancient ones (*Figure 6*). Most of the shared haplotypes belong to *L. guanicoe guanicoe* and *Lama glama* haplogroups. The temporal network suggests a loss of ancient mitochondrial lineages, mostly in the vicuña (Table S9).

Control region haplotype diversity values were relatively similar between the modern samples (0.9765) and the ancient samples (0.9348). However, given the uneven sample size between modern and ancient samples, we randomly sampled with replacement 10,000 times the modern datasets conditional on a sample size like the ancient samples. For each of the random samples, we estimated the haplotype diversity and the number of substitutions observed. We used that information to draw a distribution of those two summary statistics of genetic diversity conditional on the ancient DNA sample size. The analysis of all the data (i.e. without differentiating between large and small animals) indicated that the ancient samples do indeed present a high genetic variation given their sample size as their diversity measures fall in the 93rd and 94th percentile of the distribution of the random samples from modern animals (*Figure 7A and D*). For the large animals, we observed that the ancient samples fall in the 89th and 95th percentile of the distribution of random samples from the modern *Lama*, while for the small animals, the ancient samples fall in the 98th percentile of the distribution of random samples of *Vicugna* for both statistics.

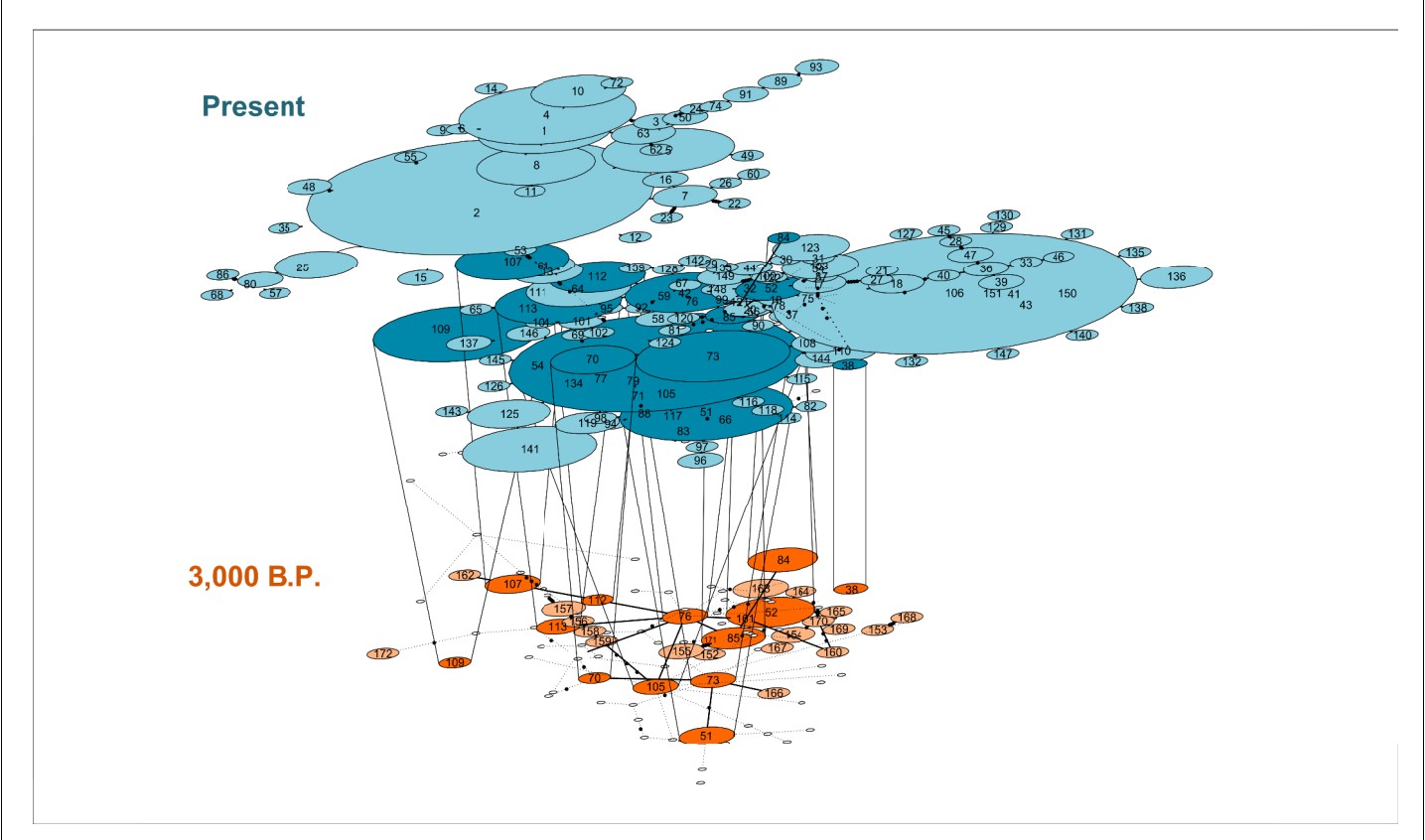

**Figure 6.** Temporal statistical parsimony haplotype networks for *Lama guanicoe, Lama glama, Vicugna vicugna,* and *Vicugna pacos* with ancient samples. Haplotypes of modern camelid sequences are colored in blue and ancient DNA haplotypes in orange. Each mutation is connected by one black circle. If two haplotypes are separated by one mutation, they are connected by one line. Haplotypes shared by modern and ancient samples are connected by vertical lines showing dark blue and dark orange color (Table S9).

## Discussion

Here, we present the most comprehensive dataset of mitogenomes from SACs to date. This includes a collection of sequences reflecting zooarchaeological samples (ancient DNA) from the San Pedro de Atacama Desert and dates between ~ 3,150 and 2,380 yr BP, and a wide collection of modern samples representing the four extant species. We used these data to compare patterns of genetic variation between taxa and across time to generate insights into hypotheses about the domestication of alpaca and llama. This dataset not only presents unexpected information about the domestication history of SACs, it also provides new insights into the loss of genetic diversity in the extant species.

Zooarchaelogy's foundation is the ability to accurately identify species from animal remains in order to develop an interpretation of human and environmental interactions within an archaeological context (*Peres, 2010*). While traditionally the field has largely relied on species identification based on comparative analyses with zoological collections or based on researcher's experience, developments in genetic techniques have facilitated comparisons even when morphological analyses are compromised (*Peres, 2010*; *LeFebre and Sharpe, 2018*). Species identification based on bone morphology is not always possible when species are closely related and only have marginally diverged with respect to each other (*Frantz et al., 2020*). However, for SACs, differentiating between the larger *Lama* animals and the smaller *Vicugna* animals is possible (*Wheeler, 1985*; *Wheeler, 1995*). The bone morphological analyses implemented here indicate that the astragalus and phalanx have a good discriminatory power to differentiate between larger (*Lama*) animals and the small (*Vicugna*) animals (*Figure 2a and b*). The statistical test between the different measurements carried out for

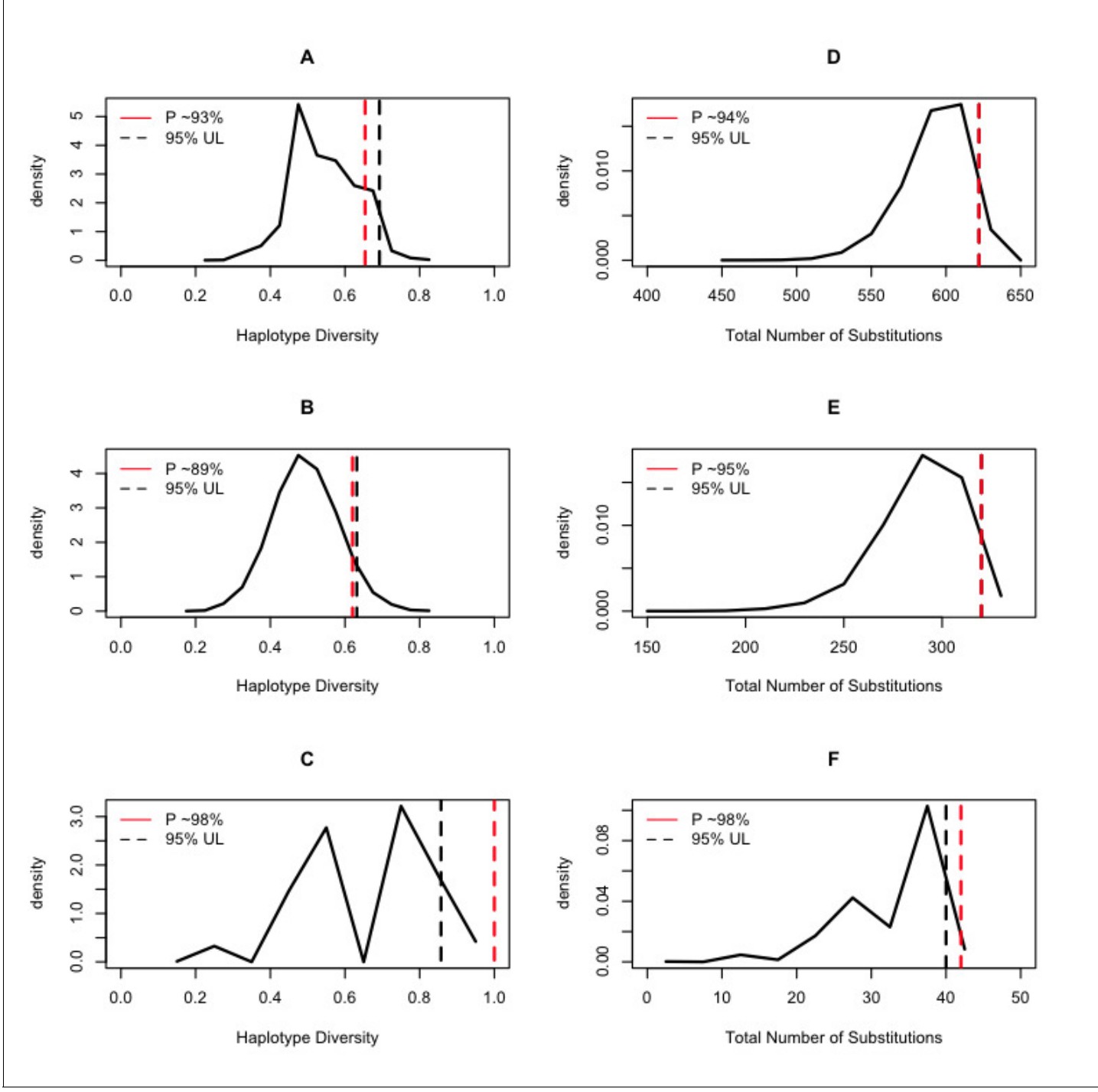

**Figure 7.** Expected distributions of summaries of genetic diversity conditional on the aDNA sample size. The column on the left shows the results of estimating the haplotype diversity on random samples of the modern animals conditional on the sampling size of the ancient DNA samples. The column on the right shows the results of counting the number of substitutions on the random samples of the modern animals conditional on the sampling size of the ancient DNA samples. The black dashed line is the 95% upper bound of the statistic, and the red dashed line is the position of the ancient DNA data summary statistics of diversity within the context of the random samples of extant samples.

each of the bones showed that the large specimens are significantly different from the small specimens (all p-values were smaller than $10^{-4}$).

The phylogeny generated using 130 ancient and modern camelid mitogenomes places the vast majority of the ancient samples on the expected clades based on bone morphology (*Figure 3*). Namely, the large sized animal bone samples were grouped with modern samples of *Lama* (95%) with the exception of two samples that grouped with the *Vicugna vicugna* clade (Clade 4). The converse also occurred with the majority of the small sized animal bone samples that clustered in Clade 4 (63%) with the exception of seven (C77, C69 C70,C1, C12 from Tulán-54 and C65, C6 from Tulán-85) that grouped with *Lama* (Clades 2 and 3).

## Domestication history of llamas and alpacas

Few domestic animals have had their origin as contested as llamas and alpacas. While their wild relatives, the guanaco and the vicuña, have been determined to have diverged from their common ancestor two to three million years ago (*Stanley et al., 1994*; *Wheeler, 1995*; *Fan et al., 2020*), the domestic llama and alpaca are assumed to have been domesticated 6,000 to 7,000 years ago (*Wheeler, 1995*). Despite this, a lack of congruence between methodological analyses has resulted in alternative domestication scenarios being put forward. If the llama were domesticated from guanaco and alpaca from vicuña, we would have expected the phylogenetic tree to group those pairs of taxa accordingly.

Our analyses place modern alpaca, with the exception of two samples, within the genus *Lama* alongside guanaco, modern llamas, and zooarchaeological samples of both large- and small sized animals. While llamas are a paraphyletic group with two main branches (Clades 1 and 3) within the greater *Lama* clade. Within Clades 1 and 3, alpaca groups within llamas supporting the hypothesis that llamas and alpacas were both domesticated from guanacos (hypothesis 3) while vicuña was possibly never domesticated.

Such a scenario would assume, however, that no alpaca should cluster with vicuñas and that the clade formed by llamas and alpacas has the guanaco as sister taxon (*Renneberg, 2008*) which is not observed in our data. Instead, our mitogenomic phylogenetic trees suggest a modification of such hypothesis where llamas derive from two separate guanaco clades. We suggest that the wild component of Clade 3 is now extinct, while the extant guanacos derive from group which Clade 1 llamas originated from. Such a scenario is consistent with the zooarchaeological and genetic evidence for multiple llama domestication centers across the Andes (*Wheeler et al., 2006*; *Kadwell et al., 2001*; *Bruford et al., 2003*), including sites in Northwestern Argentina and Northern Chile (*Yacobaccio, 2001*; *Mengoni Goñalons and Yacobaccio, 2006*; *Cartajena, 2009*; *Núñez and Calogero, 1988*; *Olivera, 2001*; *Barreta et al., 2013*) and in the Peruvian central Andes (*Mengoni Goñalons and Yacobaccio, 2006*; *Moore, 2011*).

The domestication hypothesis described above would imply that the alpaca was simultaneously domesticated alongside with the llama from the two guanaco ancestors based on the clustering of alpacas and llamas in our phylogenetic tree. While that could have happened, it is also possible that alpacas are the outcome of hybridization between llamas and vicuñas (*Wheeler, 1995*; *Wheeler, 2006*; *Renneberg, 2008*). Alpacas are assumed to have been domesticated independently of llama and probably earlier too (*Wheeler, 2016*). Peruvian archaeological time series showing the transition from human hunting of wild vicuñas and guanacos (9,000–6,000 years ago) to the establishment of early domestic forms (6,000–5,500 years ago) and the development of human societies centered around pastoralism focused on alpacas and llamas management at high altitude starting 5,500 years ago (*Wheeler, 1985*; *Wheeler, 1999*). Yet, so far, no evidence of domestication of alpacas has been found outside the domestication center in central Peru (for which no zooarchaeological or modern samples that could cluster with the vicuña samples are thus far available). However, the archaeological evidence indicates that pre-Columbian alpaca populations were substantially larger than today and that a dramatic bottleneck occurred during the Spanish conquest (*Wheeler, 2016*). Thus, it is possible that some of the genetic information about multiple domestication centers for alpaca, as observed in llama, does not exist anymore. Alternatively, the zooarchaeological samples of Tulán that are putative alpacas may represent one of those non-central Peruvian alpaca domesticated lineages resulting from the cross of female llamas with male vicuñas resulting in phenotypical alpacas that carry a *Lama* mitogenome.

## Early hybridization of domesticated camelids

Modern alpacas and llamas are the outcome of an extensive hybridization that has been attributed to the species near demise following the Spanish conquest (*Kadwell et al., 2001*; *Casey et al., 2018*; *Fan et al., 2020*), hybridization that so far has not been shown to have affected the wild SACs (i.e. guanaco and vicuña). During the Spanish conquest, camelid populations were drastically reduced (Gade, 2013), probably due to (i) a lack of understanding of the superior quality of their fiber for the textile industry, (ii) the interest in the introduction and establishment of cattle and sheep for the production of wool that was in high demand in the Low Countries, and (iii) as a measure to control local human populations by destabilizing their means of sustenance. Consequently, the paralleled decline in the Indigenous population and the disappearance of their camelid breeding tradition due to the lack of a written system resulted in a small domestic camelid population that largely was left unmanaged and which could interbreed (*Wheeler et al., 2006*; *Marín et al., 2007b*; *Pinto, 2012*). Hybridization between llamas and alpacas has become such an extensive issue that up to 40% of llamas and 80% of alpacas show evidence of hybridization (*Kadwell et al., 2001*), and through recent whole genome sequencing efforts, *Fan et al., 2020* reported that up to 36% of the modern alpaca genome is likely derived from llama hybridization.

In particular, *Fan et al., 2020* timed the hybridization history between SACs and found evidence for a postcolonial hybridization, and not an ancient one, thereby suggesting that the bulk of the taxonomic uncertainty driven by this process affects modern samples. Moreover, although rare, hybridization between llamas and guanacos (i.e. large bodied animals) and separately between alpacas and vicuñas (small bodied animals) has been previously reported (*González et al., 2020*). While the results presented here do not disagree with the hybridization patterns described elsewhere (i.e. the recent hybridization history timed by *Fan et al., 2020*), the observation of hybrid animals among the zooarchaeological samples (i.e. those samples that present a discordance between their taxon identification based on bone morphology and their molecular taxonomic identification) suggests that the hybridization history of the domestic camelids is yet far from understood, and could have also occurred prior to the European colonization of South America. Moreover, artificial hybridization (human driven) between SACs mostly occurs between crosses of one male with several females of another species (*Marín et al., 2017*), with the offspring only carrying the maternal mitochondrial DNA and thus presenting no evidence in this locus of the hybridization event from which they derive. Consequently, the results presented here advocate for efforts to be made to attempt sequencing the Y-chromosome of ancient samples that may further help unraveling this hybridization history. In the absence of Y-chromosome sequences, efforts to obtain ancient nuclear DNA sequences discriminating between the four species would also help understanding the patterns of hybridization as indicated with modern samples by *Fan et al., 2020*.

However, despite advances in the field of ancient DNA, it still remains very challenging to obtain nuclear data from ancient samples, due to the fragmentary nature of the DNA on these samples and the low endogenous content and, so far, it has not been possible to be obtained from ancient SAC samples (*Lindahl, 1993*; *Collins et al., 2002*; *Bollongino et al., 2008*; *Pruvost et al., 2007*; *Hofreiter and Shapiro, 2012*). In fact, the hybrid samples (i.e. C69, C77, and C70) present in Tulán (3,400–2,300 yr BP) not only further support the presence of domestic camelids associated with the human settlements in the region, they also provide information about breeding practices by these ancient formative pastoralist communities. Such ancient hybridization is consistent with hypotheses of human controlled crossings between species during the domestication period and afterward to maintain variation of interest (e.g. adaptive) (*Moore, 2011*; *Marshall et al., 2014*).

Furthermore, although the wild, a hybrid lineages can consistently carry the same mitochondrial haplotype throughout time, they may change body size (e.g. become larger) through backcrossing as the host population genome becomes prevalent in the lineage due to Mendelian inheritance. Lastly, it is possible that the ancient DNA samples in Clades 1–3 represent mostly guanacos and not domestic camelids, suggesting that llamas were domesticated from two different guanaco clades (i.e. Clades 1 and 3) in line with *Yacobaccio, 2004*, *Gallardo and Yacobaccio, 2005* and *Mengoni Goñalons, 2008* that suggest two different domestication centers. However, the presence of both domestic camelids in Tulán has been previously established through the analyses of bone remains (*Cartajena et al., 2007*; *Cartajena and López, 2011*; *Núñez et al., 2017b*), microscopic

camelid fiber analyses (*Benavente, 2005*; *Benavente, 2005*), and the analyses of bone collagen stable isotopes that differentiate llamas ritually fed with maize from wild guanacos (*López et al., 2013*).

Although our results support the guanaco as the ancestor of llama, we propose a more complex evolutionary history for the alpaca. As small sized animals (putatively alpaca) are found in both the *Lama* and *Vicugna* clades, alpaca may have derived in some part from both species with an important component of its genetic variation comprises guanaco, which indicates that hybridization between these species played a role in its evolutionary history.

## Loss of genetic variation

The demographic analyses of the ancient and modern samples of each clade in our tree revealed contrasting demographic histories across the dataset. The BSP inferred from the wild animals (Clades 2 and 4) showed that substantial demographic expansions started in the middle of the previous interglacial and continued expanding throughout the last glacial period. Such an observation is consistent with an increase in habitable land for *Lama* and *Vicugna* as their distribution ranges probably shifted north and toward lower altitudes as the far South and mountaintops started to freeze. Contrastingly, the ancient domestic samples from Clades 1 and 3 present a roughly stable ancient demography (Clade 3) or a comparatively moderate population expansion at the onset of the last glacial period (Clade 1). Interestingly, the four clades all show a demographic decline starting near ~ 10,000 yr BP reaching its lowest point around ~ 6,000 years ago. These dates coincide with the establishment of human populations throughout the upper Andes ~ 9,000 yr BP (*Aldenderfer, 1999*) and the establishment of a specialized hunting for vicuña and guanaco (9,000–6,000 yr BP) prior to the onset of the domestication process (*Pires-Ferreira et al., 1976*). The brief reduction in effective population size is recovered by a population expansion also present in the four clades but exacerbated on the clades with domestic animals where the recovery is substantially larger (Clades 1 and 3; *Figure 4*). The larger population expansion in Clades 1 and 3, rather than on the clades with wild animals (Clades 2 and 4), is consistent with an increase in domestic animals population size following the onset of the domestication process and the breeding of camelids with traits of interest (*Wheeler, 1995*; *Thompson et al., 2006*; *Baied and Wheeler, 1993*).

Taken together, the comparative analysis of both the demographic histories and levels of genetic diversity in both datasets indicates that the ancient camelids presented a much larger genetic pool than the modern ones. In the BSP, it can be seen that the plots obtained from the ancient samples (albeit fewer than the modern ones) result in substantially larger effective population sizes and more ancient coalescent times (*Figure 4*), reflecting higher genetic variation in those samples. Similarly, the genetic diversity measured in terms of haplotype sharing between ancient and modern samples, or the number of haplotypes and total number of substitutions observed in both datasets (*Figure 7*), also shows that the ancient samples presented significantly more genetic variation than the modern ones, with the vicuña having lost the most genetic variation across time. The latter is consistent with the evidence for bottlenecks in the species (*Wheeler, 1995*; *Casey et al., 2018*; *González et al., 2019*; *Fan et al., 2020*).

This is not surprising when considering the relative homogeneity of the environment where vicuñas are found and how gregarious they are in comparison to guanacos. Vicuñas are constrained to the upper Andes High Plateau at altitudes above 3,500 masl, while guanacos are largely freely roaming and found across a much wider altitudinal gradient from the sea level to the same altitudes where vicuñas are found. Additionally, vicuñas are gregarious animals being able to form herds of 100 animals or more in some parts of their Peruvian range (*V. v. mensalis*) where guanacos (*L. g. cacsiliensis*) tend to form small family groups of 5–10 individuals (usually one male and one to two females with their offspring), while in the southern ranges vicuña groups tend to be smaller (*V. v. vicugna*) and the guanacos (*L. g. guanicoe*) are larger than in northern locations. Such differences between the two species likely made the vicuña a preferential hunting target for the upper Andean human population, as they were easier to find in large quantities and because their groups were found in relative proximity to each other due to the environmental constraint of their habitat distribution. Lastly, by inhabiting a much more homogeneous habitat in the upper Andes (the High Plateau) reflecting the narrower gradient of the Andean slopes where they are found, vicuñas are more susceptible to environmental changes that affect their habitat as these likely impact their entire range. Nevertheless, while human-driven demographic changes are likely to have driven some of the changes observed in this data, the effect of environmental changes cannot be ignored, in particular

during the middle Holocene climatic optimum (~8.2 kyr BP–4.2 kyr BP). This time span is marked by a period of strong hydroclimate shifts that led to the aridization of the Andes as rainfalls became more stochastic and temperatures increased (*Riris and Arroyo-Kalin, 2019*). Such aridification is likely to have changed the habitable range for the vicuña, further exacerbating the demographic pressure that it was already experiencing.

Finally, all four SAC species show evidence of having lost a very large proportion of their genetic variation as modern populations only represent a small fraction of historical population diversities. This loss of diversity may lead to problems in the survival of these species in the future. Although some extant camelid populations have been brought back from the brink of extinction within the last 50 years (e.g. the Peruvian vicuña was once as few as 5,000 individuals mid last century, but through the international Plan de Manejo de la Vicuña it has since recovered to over 100,000 individuals), current climate change exacerbating the aridification of the upper Andes, land conversion for agriculture, and a general lack of appetite from local governments to invest in these native species of agricultural value are likely to drive these camelids into further trouble.

# Materials and methods

## Archaeological samples

The study area is located in the II Region of Chile (Antofagasta), specifically in the western slope of the Puna de Atacama (22°−24°S) (*Figure 1—figure supplement 1*) at the south eastern border of the Atacama saltpan. These samples were collected in the archaeological sites Tulán-54 (3,360–2,950 to 2,410–2,370 cal. yr BP; 57 samples) and Tulán-85 (3,480–3,210 to 2,410–2,370 cal yr BP; 18 samples), corresponding to the Early Formative period (*Núñez et al., 2017a*). During the Early Formative period (ca. 3,360 to 2,370 cal. yr BP) human groups underwent important cultural and economic transformations. Early pastoralist communities were sustained by hunting, gathering and camelid breeding. They started integrating an important ritual components, new technologies (e.g. pottery), the development of ceremonial and monumental architecture. There was an increasing of macro-regional interaction networks, especially between the coast and Northwestern Argentina (*Núñez et al., 2005*; *Núñez et al., 2017a*; *Núñez et al., 2017b*).

## The archaeological context

The site Tulán-54 is located on an esplanade on the southern edge of the Tulán ravine and Tulán-85 is located on the border of the Salar de Atacama salt flat at an altitude ranging from 3,200 masl to 2,300 masl in less than 30 km (*Figure 1—figure supplement 1*), enabling these groups to exploit different resources due to the altitudinal differences (*Núñez et al., 2017b*). This area presents the most complete cultural sequence of the western slope of the puna, which begins at 12,725–11,995 cal. yr BP (*Loyola et al., 2019*) until recent times, as well as a well-established paleoclimatic framework (*Núñez et al., 2002*). Tulán-54 is formed by an extensive mound (2,800 m2), which overlays several areas of accumulation and structures. A sunken ceremonial structure was identified at the center that had been filled over time covering the entire structure, forming a stratified mound (*Figure 1—figure supplement 2*). The structure is surrounded by a perimeter wall and the inner space is divided into six precincts, where 27 burial pits with human infant remains and fireplace structures were found. A smaller contemporary wall was identified neighboring the central semi-subterranean ceremonial structure, revealing a complex ceremonial architectural layout. Several additional areas with accumulations and underlying structures were found outside the temple along with rudimentary stone structures related to nonpermanent domestic activities (*Núñez et al., 2017b*; *Cartajena et al., 2019*).

## Sample collection

Samples from Tulán-54 were selected from inside and outside of the main and smaller ceremonial structure and outside (35 phalanx and 22 astragalus) in order to cover different sections of the site (Table S1). Tulán-85 corresponds to an extended mounded stratified deposit (2,500 m2), interpreted as an area of domestic activity associated to human newborn burials and few domestic structures, however no ceremonial architecture was found (*Figure 1—figure supplement 2*). Samples were collected from the activity area and structures (16 phalanx and two astragalus, Table S1). We

additionally collected two samples from the sites Tulán-52 (5,290–4,840 to 4,430–4,090 cal. yr BP) and Tulán-94 (3,715–3,560 to 3,460–3,200 cal. yr BP), belonging to the Late Archaic and Transitional Archaic-Formative phase, respectively, to evaluate the earlier presence of domestic animals due to the Late Archaic domestication process (*Cartajena et al., 2007*; *Cartajena, 2013*) (Table S1).

## Morphometric analysis

We recorded osteometric measurements from 61 out of the 77 bone samples from the archaeological sites using Vernier calipers to the nearest 0.1 mm, as described by *Driesch, 1976*. Due to fragmentation or bad preservation, 18 out of 61 samples were not used for morphometric analysis. Osteometric measures were performed on each bone sample prior to drilling for aDNA to avoid bias due to damage during drilling. Measurements were performed on phalanx and astragalus. Bones were selected from different levels and squares in the excavation sites in order to choose different individuals (Table S4). Only adult individuals determined by a phalanx with fused epiphysis were selected for the morphometric analysis. The first anterior phalanx was selected due to its high frequency in the sites and because it is easy to differentiate between anterior and posterior phalanx. Breadth of proximal articulation (BFp) and the depth of the proximal epiphysis (Dp) measures were taken from the proximal side of the phalanx. In the case of the astragalus, growth occurs from a single primary center of ossification, with the bone continuing to grow after birth until the bone is completely ossified. The reduction of bone porosity and well-delimited medial intertarsal and plantar trochlea articular surfaces were used as criteria to identify bones from adult animals that were collected for this work. Modern measurement of big sized subadult individual camelids does not overlap in size with those of adults in the small group (*Cartajena, 2003*); however, although unlikely, we cannot rule out the possible overlapping due to age in the smaller size camelid group (*Izeta, 2004*). Astragalus measures included the breadth distal (Bd) and greatest length medial (GLm) (*Driesch, 1976*). Morphological differentiation between 'small size animals' and 'big size animals' was tested using correlations between the anterior and posterior phalanx and astragalus using the program PAST (*Hammer et al., 2001*). In order to increase the sample size and robustness of our analyses, our data were combined with modern published morphological data the four modern camelid species (*Cartajena, 2003*; *L' Heureux, 2010*).

## Ancient DNA extraction

DNA was extracted from 61 samples of phalanx and astragalus bones in a dedicated ancient DNA laboratory at the GLOBE institute, University of Copenhagen. From each specimen, 150 mg of bone powder was drilled using a Dremel drill to obtain bone powder for DNA extraction. DNA extractions were performed as in *Dabney et al., 2013* with the following modifications: samples were incubated overnight with the extraction buffer at 42°C instead of at 37°C, the bone powder was pelleted out of suspension, and the supernatant concentrated down to 150–200 µl for each sample using 30 kDa Amicon centrifugal filter unit (Milipore). Binding buffer 13 times larger in volume was added to the concentrated supernatant and DNA was purified with MinElute columns (Qiagen) following the manufacturer's instructions with the exception of a 15 min incubation at 37°C during the elution step. DNA was eluted by adding 50 µl of EB buffer twice for a total volume of 100 µl, in order to increase DNA yields. For every round of extractions, one blank extraction control was used to monitor for contamination.

## Ancient DNA library build

Double stranded libraries were built using 21.25 µl of the extracts following the protocol from *Meyer and Kircher, 2010* with some modifications: the initial DNA fragmentation was not performed and MinElute kit (Qiagen) was used for the purification steps. A quantitative PCR (qPCR) assay was performed prior to the amplification step to assess the optimal number of cycles for index PCR to avoid overamplification using Roche LC480 SYBR Green I Master Mix with the primers IS5 and IS6 from *Meyer and Kircher, 2010*. All qPCR were carried out in a total volume of 20 µl with the following reaction conditions: 95°C/10 min, 35 cycles of 95°C/30 s, 55°C/30 s, 72°C/30 s followed by 1 cycle at 95°C/30 s, 55°C/1 min and a detection final step at 95°C/30 s. Indexing PCRs were performed in 50 µl reactions using 25 µl KAPA HiFi HotStart Uracil+ReadyMix, 2 µl of forward InPE 1.0 (10 µM) (*Meyer and Kircher, 2010*), 2 µl of an index primer containing a unique 6-nucleotide index

tag (10 Mm), and 21 µl of DNA library. The cycling conditions of amplification PCR were 45 s at 98°C, 14–17 cycles (at 98°C for 15 s, 65°C for 30 s, 72°C for 30 s) and a final elongation at 72°C for 1 min. MinElute columns were used for purification after index PCR and PCR products were quantified using a Qubit Fluorometer (HS). In order to reach the required 300–500 ng of starting material for performing the capture experiments, some of the libraries were reamplified. A re-amplification PCR was setup with 10.5 µl of the indexed library, 12.5 µl of KAPA HiFi HotStart Uracil+ReadyMix, and 1 µl of each re-amplification primer, IS5 and IS6 (10 µM stock concentration) (*Meyer and Kircher, 2010*). Cycling conditions were the same as for the index PCR but seven cycles for annealing. PCR products were purified and quantified as indicated above.

### Ancient mitochondrial DNA capture

Our sequencing libraries underwent target-capture prior to sequencing in order to enrich our libraries of endogenous mitochondrial DNA sequences. We designed a camelid set bait which was composed of 120 nucleotides baits tiled every four bases across a representative sample of four living camelid mitochondrial genome sequences. The 6234 baits were then synthesized at MYcroarray (https://arborbiosci.com/) as part of several MYbaits targeted enrichment kits. Hybridization by capture was performed following the MYbaits v3 manual, using 350–500 ng of library starting material. We performed a single round of enrichment at 55°C for 24 hr. Enriched libraries were eluted in a total volume of 30 µl. Post-capture amplification was carried out according to the suggested protocol using KAPA HiFi HotStart Uracil+ReadyMix and 12 cycles for the cycling PCR conditions. Enriched libraries were quantified using Agilent 2200 TapeStation System, pooled in equimolar amounts, and sequenced on an Illumina HiSeq 2500 using 80 bp single-end chemistry at the Danish National High-Throughput Sequencing Center.

### Ancient DNA bioinformatic processing

PALEOMIX was used to perform basic read processing (*Schubert et al., 2014*): (i) adapter trimming, (ii) mapping trimmed reads to the reference mitogenome of *Vicugna pacos* (GenBank: Y19184.1), and (iii) PCR duplicate removal. Seeding was disabled for mapping and low-quality reads of less than 30 bp were discarded. The BAM files with unique reads were loaded into Geneious v7 (*Kearse et al., 2012*) to generate mitochondrial consensus sequences for each sample. Mitochondrial consensus sequences were generated using the 'strict' consensus that calls bases using majority rule and requires a minimum depth-of-coverage of 3x (Table S5). This resulted in the generation of 61 complete mitochondrial genomes. DNA damage and fragmentation patterns were also estimated within the PALEOMIX pipeline using MapDamage2 (*Jónsson et al., 2013*; *Figure 3—figure supplement 1*).

### Dataset of new modern DNA samples

For comparison with the aDNA samples, blood and tissue samples were obtained from 66 georeferenced modern samples to generate complete mitochondrial genomes excluding the control region (15438 bp; *Figure 1—figure supplement 1 and 3*). These include 16 guanacos: Northern guanaco (6 *L. guanicoe cacsilensis*) and Southern guanaco (10 *L. guanicoe guanicoe*); 17 vicuñas: Northern vicuña (6 *V. vicugna mensalis*) and Southern vicuña (11 *V. vicugna vicugna*) from throughout the current distribution across Peru, Argentina and Chile; 16 llamas from a slaughterhouse in Putre, Chile; and 17 alpacas from a farmer and breeder in Chile. Genomic DNA was extracted using the Puregene Tissue Core Kit A (Qiagen). For each individual, 1–3 µg of DNA was sheared into fragments of 150–700 bp prior to the construction of the sequencing library. Adapter ligation (or library build) was performed using the NNNPaired-end sequencing library kit with an insert size of approximately 300 bp. Libraries were sequenced to mid/low-coverage (11-14x) on an Illumina Hiseq X Ten platform.

From the resultant raw reads, we removed reads with $\geq$ 10% unidentified nucleotides, >10 nucleotides aligned to the adaptor, with $\leq$ 10% mismatches allowed, with > 50% bases having phred quality < 5 and putative PCR duplicates generated during library construction. Following the initial processing, unpaired reads were removed using Trimmomatic (*Bolger et al., 2014*), and reads with a Phred quality score lower than 33. High-quality reads were aligned to conspecific reference mitochondrial genomes (access number NCBI, EU681954; FJ456892; AJ566364; AP003426) as appropriate using BWA (*Li et al., 2009*; *Li, 2013*), the resultant Bayesian alignments were sorted and filtered

using SAMtools program (*Li et al., 2009*). The latter commands included filters to eliminate optical duplicates, unmapped reads and mates, and only keeping reads that mapped completely. We reconstructed the mitochondrial consensus sequences using GenomeAnalysisTK.jar from GATK with references indexed with Picard (*Li, 2014*).

## Mitochondrial sequences compilation

We included 300 bp mitochondrial control region sequences from 815 modern camelids representing the Northern guanaco (54 *L. guanicoe cacsilensis*), the Southern guanaco (366 *L. guanicoe guanicoe*), the Northern vicuña (218 *V. vicugna mensalis*), the Southern vicuña (35 *V. vicugna vicugna*), domestic alpaca (53 *V. pacos*), domestic llama (79 *L. glama*) and 10 hybrids (three hybrids between the two vicuña forms and seven hybrids between llama and guanaco; Table S2). Additionally, three ancient guanaco mitogenomes from Isla Mocha (Chile) (KX388532.1; KX388533.1; KX388534.1), one guanaco (NC_011822.1), one vicuña mitogenome (FJ456892.1), one llama mitogenome (AP003426.1) and two alpaca mitogenomes (KU168760.1; AJ566364.1) were downloaded from NCBI and included in the analyses (Table S3).

## Phylogenetic and demographic analysis using mitogenome data

The 131 sequences from ancient and modern SAC mitogenomes and one Bactrian camel sequence used as outgroup, were aligned using Geneious V7, resulting in an alignment of 15,456 bp, including protein-coding genes, transfer RNA (tRNA), ribosomal RNA (rRNA) genes, and a partial control region sequence (CR). Partition Finder v1.1.1 (*Lanfear et al., 2012*) was used to determine the optimal partitioning scheme for this alignment and substitution model for downstream analyses. Several partition schemes were tested (Table S6). The best fitting-model to our alignment consisted of five partitions (tRNA and rRNA as one partition each and protein-coding genes divided by codon as three separate partitions) with HKY+I+G for partitions 1, 2, 3, and 5 and GTR+I+G for partition 4 as substitution model.

Phylogenetic relationships were estimated using the software MrBayes v3.2.6 (*Ronquist and Huelsenbeck, 2003*) and the substitution evolution models identified above. The MrBayes MCMC algorithm was run twice with three cold and one hot chain for $1 \times 10^6$ generations, sampling every $1 \times 10^3$ generations and discarding the first 25% of steps of the MCMC as burn-in following visual examination of the stabilization of the likelihood values of the inferred tree. Trees were summarized with the majority-rule consensus approach, using posterior probability as a measure of clade support. The final consensus tree was visualized in FigTree v.1.4.2 (*Rambaut, 2014*). The number of distinct haplotypes, haplotype diversity, and nucleotide diversity were estimated in R using pegas v0.12 (*Paradis, 2010*) and ape v5.3 (*Paradis and Schliep, 2019*). We estimated nucleotide diversity excluding sites with gaps and missing data in each pairwise comparison.

Additionally, ultrametric trees were estimated for the dataset using BEAST v.1.8.0 (*Drummond et al., 2012*) under a coalescent model and with *C. dromedarius* (NC_009849.1) and *C. bactrianus* (NC_009628.2) as outgroups. A Shimodaira-Hasegawa test for the evolution of the trio vicuña, guanaco, and dromedary was calculated in MEGA X (Molecular Evolutionary Genetics Analysis across computing platforms) (*Kumar et al., 2018*) and showed no significant evidence of deviation of the data from the molecular clock null hypothesis. Thus, the number of observed substitution differences between the pair vicuña/dromedary (guanaco/dromedary had the same value) were used to estimate a substitution rate of 2.28-e8 per million years (based on timetree.org [*Kumar et al., 2017*] divergence of 20.56 million years reported between these two taxa) for the BEAST analysis. A total of 20 million steps of BEAST's MCMC algorithm were carried out discarding the first 10% as burn-in. Stability of the BEAST run was determined by achieving ESS (Effective Sample Size) values larger than 200 which were visualized in Tracer (*Rambaut et al., 2018*) and convergence was determined by running the analysis in duplicate and obtaining the same tree. To investigate changes of population size through time, a Bayesian Skyline reconstruction was performed on the data using BEAST and visualized in Tracer v1.4.1 (*Rambaut et al., 2018*). The plot was constructed with a rate of base substitution of 1,2% substitution per base pair per million years and HKY as a model of nucleotide substitution with an MCMC of 10 million steps following a discarded first 10% of each chain as a burn-in.

### Haplotype networks using mitochondrial control regions sequences

To explore the relationship between the different subspecies and their wild relatives, we aligned the partial CR (control region) from the ancient and modern mitogenomes with a total of 849 specimens for the four species covering the distribution range across Peru, Chile, Argentina, and Ecuador using MAFFT (*Katoh and Standley, 2013*) (Table S2). The final alignment consisted of 300 bp from the hypervariable I domain. We used this dataset to calculate a haplotype median-joining network using PopART (*Leigh and Bryant, 2015*) and to investigate the change in haplotype composition over time using TempNet (*Prost and Anderson, 2011*) by dividing the dataset into two time periods, that is, the 41 ancient samples and the 808 modern samples.

## Additional information

### Funding

| Funder | Grant reference number | Author |
|---|---|---|
| Fondo Nacional de Desarrollo Científico, Tecnológico y de Innovación Tecnológica | 1070040 | Paloma Diaz-Maroto Isabel Cartajena Lautaro Núñez |
| Fondo Nacional de Desarrollo Científico, Tecnológico y de Innovación Tecnológica | 1020316 | Paloma Diaz-Maroto Isabel Cartajena Lautaro Núñez |
| Fondo Nacional de Desarrollo Científico y Tecnológico | 1130917 | Paloma Diaz-Maroto Isabel Cartajena Lautaro Núñez |
| Fondo Nacional de Desarrollo Científico y Tecnológico | **1140785** | Juan Carlos Marin |

The funders had no role in study design, data collection and interpretation, or the decision to submit the work for publication.

### Author contributions

Paloma Diaz-Maroto, Conceptualization, Resources, Data curation, Software, Formal analysis, Funding acquisition, Validation, Investigation, Visualization, Methodology, Writing - original draft, Project administration, Writing - review and editing; Alba Rey-Iglesia, Software, Supervision, Methodology, Writing - original draft; Isabel Cartajena, Conceptualization, Supervision, Investigation, Project administration, Writing - review and editing; Lautaro Núñez, Conceptualization, Supervision, Validation, Writing - review and editing; Michael V Westbury, Software, Formal analysis, Writing - review and editing; Valeria Varas, Software, Visualization, Methodology; Mauricio Moraga, Formal analysis, Visualization, Methodology; Paula F Campos, Conceptualization, Investigation, Methodology; Pablo Orozco-terWengel, Software, Supervision, Visualization, Methodology, Writing - review and editing; Juan Carlos Marin, Resources, Data curation, Software, Supervision, Validation, Methodology, Writing - review and editing; Anders J Hansen, Supervision, Funding acquisition, Validation, Investigation, Project administration, Writing - review and editing

### Author ORCIDs

Paloma Diaz-Maroto (iD) https://orcid.org/0000-0002-3937-6609
Paula F Campos (iD) http://orcid.org/0000-0003-1285-4671
Pablo Orozco-terWengel (iD) https://orcid.org/0000-0002-7951-4148
Juan Carlos Marin (iD) https://orcid.org/0000-0002-5696-9422

### Decision letter and Author response
Decision letter https://doi.org/10.7554/eLife.63390.sa1
Author response https://doi.org/10.7554/eLife.63390.sa2

## Additional files

### Supplementary files
- Supplementary file 1. Supplementary tables that support the analysis and results above.
- Supplementary file 2. GenBank dataset information.
- Transparent reporting form

### Data availability
Sequencing data have been deposited in GenBank. See Supplementary file 2 for full details.

The following previously published dataset was used:

| Author(s) | Year | Dataset title | Dataset URL | Database and Identifier |
|---|---|---|---|---|
| L'Heureux GL | 2010 | Morfometría de camélidos sudamericanos modernos. La variabilidad morfológica y la diversidad taxonómica | http://www.repositorio.cenpat-conicet.gob.ar/bitstream/handle/123456789/748/zooar-queologiaAprincipiosSi-gloXXI.pdf | ISBN, 978-987-25159-6-6 |

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
