## [Decision Letter]

**Acceptance summary:**

This manuscript combines osteometric and ancient genetic data to address the persistent problem of camelid domestication in South America. It offers a large dataset of modern and ancient mitochondrial sequences from across South America and from the sites of Tulán-54 and Tulán-85 in northern Chile, respectively. By drawing samples from a time before recent major losses of camelid diversity, ancient DNA allows identification of lost lineages, and reveals layers of complexity in the process of domestication for both large and small camelid taxa. This has implications ranging from understanding the history of culturally iconic species to aiding in the conservation of threatened lineages.

**Decision letter after peer review:**

Thank you for submitting your article "Ancient DNA reveals the lost domestication history of South American camelids in Northern Chile and across the Andes" for consideration by *eLife*. Your article has been reviewed by four peer reviewers, including Jessica C Thompson as the Reviewing Editor and Reviewer #1, and the evaluation has been overseen by George Perry as the Senior Editor. The following individuals involved in review of your submission have agreed to reveal their identity: Mariana Mondini (Reviewer #3); Pamela A. Burger (Reviewer #4).

The reviewers have discussed the reviews with one another and the Reviewing Editor has drafted this decision to help you prepare a revised submission.

Summary:

Reviewers included area specialists in zooarchaeology and in genetics. All were in agreement that this is a well-written paper, with important implications for understanding camelid domestication in South America. All reviewers also agreed that the data are robust and well-presented. The content is appropriate for publication in *eLife*, and the approach is both well-justified and a welcome connection between morphology and aDNA. The reviewers focus on eight major points that should be addressed in a revision. Special attention should be given to the points about the figures, which could be made more clear.

The reviewers also wanted to make it clear that future work should involve samples drawn from a larger time span, or with more specific assignment within the ~800 year time span of the sites that dominate this sample. Specifically, it would be very useful to include earlier samples, before culturally induced interbreeding took place. Also, it would be crucial to include camelid fibers, especially those of the early type resembling modern llamas that is no longer found in extant wild populations, as they might correspond to the now extinct wild component of Clade 3. As the archaeological sample is from an area far from the current known range of alpacas – in wetter altiplano areas –, specimens confidently attributed to ancient alpacas that are not suspected to be hybrids and with less interbreeding than modern ones should also be compared in future. It is important that further studies also discuss ways to test the hypothesis of the cross of female llamas with male vicuñas to end up in "phenotypical alpacas" in San Pedro de Atacama ca. 3000 years ago.

Essential revisions:

1) Osteological measurements: There are issues here of both nomenclature and method:

– Astragali have no lines of fusion and therefore an intermediate or small size can simply be a juvenile of a larger species. No criteria are mentioned for defining the condition of "adult" for the astragali. An astragalus of a young individual of the large group could overlap in size with that of an adult of the small group. If this were to occur, some specimens could be wrongly attributed to a different size group and this, as the authors have pointed out, has implications for the taxonomic discussion. Phalanges can be more useful in this way, but they do not seem to discriminate as well. Given the history of hybridization revealed by the mtDNA analysis, there needs to be more nuance in interpreting metric data from phalanges.

– The bivariate plot of Figure 2A based on measurements of the phalanxes show two size clusters (big and small) but there are several cases (nearly a 25% of the total) that are intermediate in size. This pattern needs a further explanation that it is not included in that section of the paper. Also Figure 2B based on the astragali shows several outliers that cannot be interpreted in a straightforward manner.

– While osteometry is mentioned, there are also geometric morphometric studies that address shape instead of size to determine camelid taxa that could be mentioned. The section called "Morphometric analysis" should in fact be renamed as "Osteometric analysis," as it does not address shape.

– Figures need to be adjusted so that the samples are clearer (especially the older sample), and where the samples came from (and modern distributions) are better-justified.

2) Archaeological context of the samples: This needs to be much clearer and more specific, and there seem to be superfluous images of some contexts (like the burial) without explanation about their relevance to the sample. Specifically:

– There is little discussion about what happened with the single specimen that was older, from TU-52, or the significance/insight of the contemporaneous specimen from TU-94, which was ostensibly sampled to develop the more regional view. It is really challenging to find these on the various figures without a lot of flipping back and forth between sections of the manuscript. Information provided on Tulán-52 and Tulán-94 sites is too short and uneven. The samples from these sites (2 from each?) should better be listed in Table S1 in Supplementary file 1. It is not clear why only two samples were taken from these two sites, especially Tulán-52, which represents an earlier period that was critical in SAC domestication. In presenting the region, note that mention of San Pedro de Atacama desert only appears in the Discussion.

– What is the specific significance of the locality of Tulán with respect to this problem? How might the story change if a different locality had been sampled? The supplementary information on this is not very detailed and, as mentioned above, the significance of specimens from multiple sites at Tulán is not really explored.

– What was the context of the sampled remains? They were inside or outside of ceremonial or inhabited areas, but were they from ceremonial contexts, or trash middens? Supplementary Figure 2B – what is the purpose of illustrating the human burial with the offering? Is it because this is the context from which the camelid remains derived?

– Supplementary Figures 1 and 2: A "partial" (Tulán 54) and "panoramic" (Tulán 84) site views are provided. However, it would be useful to include a plan of both sites to know the spatial provenience of the bones discussed at a more detailed scale.

– In the subsection “Biogeographical area”, different dates are provided for the Formative period; it would be clearer to state that it starts at c. 5000 BP and the period represented in these particular sites is c. 3360-2370 BP. The inference that human populations started selecting wild camelids to raise their own herds c. 3000 years ago should be supported with some references.

3) Explicit discussion of limitations: The authors should be certain to clearly state the caveats of this study. They should be clear about the potential influence of factors, and their timing, that are independent of domestication on both size and molecular diversity. Specifically, these are limitations imposed by:

– The osteometric and aDNA methods.

– The archaeological sample relative to the research question (e.g. the chronological and geographical representation of the selected sample).

– Hybridization and later killing-off events.

– Potential impact of climatic/environmental changes.

– The extent to which any of the above is knowable.

4) Interpretations of hybridization:

In the subsection “Early Hybridization of domesticated camelids”, if modern genomes are extensively hybridized in the recent past, how valid are the modern samples used in this analysis? In other words how can a molecular taxonomic identification actually work – and be compared against the morphological results – if the modern reference collection of, say, a "guanaco" might actually have been hybridized in the recent past? And if only mitogenomes are used in this analysis, how can such hybridization truly be pulled apart? Then, the specific "hybrid specimens" described here are not really directly addressed in the Results sections; on what basis are these individuals determined to be hybrids? If it is simply the difference between morphology and mitogenome haplotype (which seems to be the case), then we should investigate which specimens were first phalanges and which ones were astragali, because the astragali seem to be much more effective at separating them out (see point #1). On the trees that code the specimens by size, I am reading this to mean that anything in the "intermediate" category was a phalanx, and therefore I'm not sure that this is a "hybrid" so much as a problem with the discrimination of the method. Alternatively, if they are truly hybrids, then there might not be expected to be a clear morphological result from the bone measurements.

5) Discussion of implications of bottlenecks:

– It is interesting that the vicuña, which according to the model reconstructed here was not a major source of domesticated stock in the past, is the one that has undergone the most genetic bottlenecking; could this potentially be as a result of human manipulation of other taxa that caused their populations to increase at the expense of wild vicuña habitat? What is the actual mechanism that the authors propose for the bottleneck in vicuña?

– The Holocene SAC demographic decline is related to human behavior in the manuscript. It would be interesting to assess the impact of climatic and concomitant environmental changes since the late Pleistocene – including the Antarctic Cold Reversal and the Hypsithermal – on their demography as well. The reasons for vicuñas undergoing bottlenecks unlike guanacos is also worth further exploring. It will also be interesting to explore why vicuña demography has recovered so nicely in recent years in spite of that, while Andean and Chaco guanacos – excluding Fuego-Patagonian steppe populations – are now so patchily distributed that are considered to be endangered. It would be interesting to know how genetic variation corresponds to camelid subspecies in modern samples, and whether this variation can be traced back in the fossil samples.

6) Adjustments to figures:

– Figure 1: The map legend should stress on what kind of data it is based, whether on the documented provenience localities where the subspecies have been previously described from a taxonomic point of view, or from any other source of information. It would be also helpful to have some precisions on the time period represented: pre or postcolonial times. The sources of the distributional information should be cited. More importantly, the fact that there are these subspecies and their main differences should be introduced when presenting SACs and before the figure. The color-coding used for the map is not enough clear to distinguish the overlapping distribution of these four subspecies, as is the case of the southern vicugna and the guanaco. Also, there are extant guanaco populations in Córdoba, Santiago del Estero and Buenos Aires provinces in Argentina that do not appear in Figure 1. The map should show the georeferenced points from where the specimens used in the analysis (data from Table S3 in Supplementary file 1). The apparent attribution to the L. g. guanicoe subspecies in the map is a supposition since the guanaco of Northwest Argentina has not been studied genetically yet. As *L.g. cacsilensis* is present in southern latitudes (for example, at Ovalle and Paposo in Chile, according to Table S3 in Supplementary file 1), distant from its northern type locality, and the Andes appear "to offer little impediment to movement" (González et al., 2006: 170) it would be important to establish if in the past there has been an interconnection between both guanaco populations.

– Figure 2: In the section "Morphometric analysis" it says that measurements were taken from 61 specimens out of a sample of 75 bones, and that of those measured specimens 18 were not considered for morphometric analysis. In a previous section it says that the mitogenomes dataset comprises 60 ancient records. These numbers do not add up. This seems to suggest that DNA was extracted from bones that were not included in the morphometric analysis. If this is true it should be mentioned. It would give a better understanding of the complementary nature of the osteometric and genetic database and the research design. The dots are also difficult to differentiate, because the labels overlap so much and the dots are the same size and color; which ones are archaeological and included versus not included in the analysis? There also seem to be no modern camelid measurements for the astragalus?

– Figure 3: Even with a zoom 200% it is impossible to read the posterior probability values of the support of the nodes. It is interesting that there are a number of "small-sized" archaeological specimens in Clade 2. Are these samples on astragali (better indicators) or phalanges (more variation)? There is a lot more variation in the sizes of adult phalanges and astragali in the archaeological sample than in the modern sample, and potentially this may have something to do with active moments of admixture and hybridization between the different clades. This relationship between the morphological data from the two different elements and how this would be represented under various hybridization scenarios could be better-addressed in the text (see point #4).

– Figure 5 is really interesting and easy to read, but surprising that the small-sized archaeological samples group exclusively away from the vicuña.

– Figure 6: Is there some way to indicate which taxa are associated with which haplogroups here, as well as linking the past with the present? It is a really useful figure, but difficult to understand the relationships between taxa (even if only the modern taxa are indicated). Perhaps the (lost) vicuña haplotypes be indicated by A vicuña figure, similar to Figure 5 – which is very clear.

– Figure 7: the labelled panels (A, B, etc.) do not have any specific information about the samples to which they pertain.

– Supplementary Figure 3: A minor thing maybe, but the underlying grey map seems to have different square shades of grey. This is a bit irritating. The legend next to the map is difficult to read, the quality of the letters is not good and the green and blue dots have grey squares around them.

7) References:

Check for missing references, incomplete ones and errors in them. There are some inconsistencies in how the authors are listed in the text and in the Bibliography, particularly with authors who have two last names/surnames. For example: "López, P." and "López Mendoza, P." are the same person; while "Mengoni-Goñalons" correctly in the text, is only "Goñalons" in the Bibliography. The authors might also consider how their names are listed here with respect to how they have published their names in other papers.

8) Formatting:

– Although it is possible to place Materials and methods to be near the front of the manuscript, *eLife* style does typically place it at the end.

– The supplementary information should also appear together at the end of the manuscript, not divided up into sections within the main body of the manuscript.

– Check for typos and formatting errors; date ranges should always be provided from the older to the younger age. Also check for grammar and sentence structure across the text, as many phrases need rewriting (for instance, that in the Abstract reading "we generated mitochondrial genomes for… 815 extant mitochondrial control region sequences from across South America"; "Northeastern Argentina" should read "Northwestern Argentina," etc.). The text needs extensive revision in this regard.

– Figures and tables are not numbered after the order in which they appear in the text, and formatting is not even (e.g., way to name pictures within one figure). While the text mentions that first phalanges were measured, the figures and tables should also read so (and not just "phalanx"). Letters in the pictures making up Supplementary Figure 2 do not match the legend; they mix with Supplementary Figure 1. Figure 3—figure supplement 3 is only mentioned in the legend of another figure and not in the main text.

---

## [Author Response]

Essential revisions:1) Osteological measurements: There are issues here of both nomenclature and method:– Astragali have no lines of fusion and therefore an intermediate or small size can simply be a juvenile of a larger species. No criteria are mentioned for defining the condition of "adult" for the astragali. An astragalus of a young individual of the large group could overlap in size with that of an adult of the small group. If this were to occur, some specimens could be wrongly attributed to a different size group and this, as the authors have pointed out, has implications for the taxonomic discussion. Phalanges can be more useful in this way, but they do not seem to discriminate as well. Given the history of hybridization revealed by the mtDNA analysis, there needs to be more nuance in interpreting metric data from phalanges.

In the case of the astragalus, growth occurs from a single primary center of ossification, after birth the bone continues growing until the bone is completely ossified. The reduction of bone porosity and well delimited medial intertarsal and plantar trochlear articular surfaces were used as criteria to select the samples. Modern measurement of subadult individuals of the big size camelid does not overlap in size with that of an adult of the small group (Cartajena, 2003). However, we cannot rule out the possible overlapping due to age in the smaller size camelid group (Izeta, 2004).

–The bivariate plot of Figure 2A based on measurements of the phalanxes show two size clusters (big and small) but there are several cases (nearly a 25% of the total) that are intermediate in size. This pattern needs a further explanation that it is not included in that section of the paper. Also Figure 2B based on the astragali shows several outliers that cannot be interpreted in a straightforward manner.– While osteometry is mentioned, there are also geometric morphometric studies that address shape instead of size to determine camelid taxa that could be mentioned. The section called "Morphometric analysis" should in fact be renamed as "Osteometric analysis," as it does not address shape.– Figures need to be adjusted so that the samples are clearer (especially the older sample), and where the samples came from (and modern distributions) are better-justified.

The provenance of the ancient samples used in this study are included in Table S1 in Supplementary file 1. You can find more information about the measurements of each sample in Table S4 in Supplementary file 1.

About the modern samples, Figure 1 has been adjusted to have a better visualization of the distribution and sampling of the modern samples used in this manuscript.

2) Archaeological context of the samples: This needs to be much clearer and more specific, and there seem to be superfluous images of some contexts (like the burial) without explanation about their relevance to the sample. Specifically:– There is little discussion about what happened with the single specimen that was older, from TU-52, or the significance/insight of the contemporaneous specimen from TU-94, which was ostensibly sampled to develop the more regional view. It is really challenging to find these on the various figures without a lot of flipping back and forth between sections of the manuscript. Information provided on Tulán-52 and Tulán-94 sites is too short and uneven. The samples from these sites (2 from each?) should better be listed in Table S1 in Supplementary file 1. It is not clear why only two samples were taken from these two sites, especially Tulán-52, which represents an earlier period that was critical in SAC domestication. In presenting the region, note that mention of San Pedro de Atacama desert only appears in the Discussion.

We mention briefly in the main text, subsection “Archaeological samples” the presence of domestic camelids during the Late Archaic period suggested as well by osteometry and fibers analysis (Cartajena et al., 2007, Cartajena, 2013). Early Formative groups show strong continuity with Late Archaic ones expressed in architectural patterns and lithic material among others (Núñez et al., 2006). Introduction.

The principal focus of this project was the material from the sites TU-54 and TU-85. During the experimental work we included two samples from TU-52 and TU-94 to evaluate the DNA preservation of material from other archaeological sites from the same area for further genetic studies at Tulán Ravine. We agree with the reviewer that it would have been a good idea to further sample other sites. However, this was not possible at the time this project was conducted.

The study area located in the western slope of the Puna de Atacama (22°-24°S) at the southeastern border of the Salar de Atacama salt is mentioned at the beginning of Materials and methods.

– What is the specific significance of the locality of Tulán with respect to this problem? How might the story change if a different locality had been sampled? The supplementary information on this is not very detailed and, as mentioned above, the significance of specimens from multiple sites at Tulán is not really explored.

We develop the description of the main formative sites (Tu-54 and Tu-85) in the text and only leave the figures in the supplementary material. As we pointed out in the main text, the Tulán ravine contains a complete occupational sequence from the Early Holocene to the recent times (modern pastoralists). It is characterized by a redundancy of occupations in a long span of time and Tu-54 is the only site with monumental architecture in the region during the Early Formative. This site could be considered as a local political and religious center. Furthermore, it shows the integration of macro-regional interaction networks between the cost and Northwest Argentina (Núñez et al., 2017a and b). These are the reasons why we chose this area to explore the history of domestication of South American camelids through ancient DNA.

Discussion.

– What was the context of the sampled remains? They were inside or outside of ceremonial or inhabited areas, but were they from ceremonial contexts, or trash middens? Supplementary Figure 2B – What is the purpose of illustrating the human burial with the offering? Is it because this is the context from which the camelid remains derived?

Tulán-54 has been defined as a ceremonial center, where the sunken ceremonial structures had been filled with remains deposited over time to cover the entire structure, forming a stratified mound made mostly of camelid bones, although lithic, ceramic, and plant remains, among others, are also present. Stratigraphic analyses showed recurrent discarding events during the occupation of the site, suggesting the practice was repeated over time until it was completely covered to a maximum depth of around 2m Cartajena et al., 2019. Samples were collected at different levels and sectors in the sites, incorporating samples from inside the ceremonial structures and outside, related to domestic activities.

The image with the human burial has been removed, it was used to illustrate the ritual context of the site, but was not in direct relation with the samples. The provenance of the samples was included in a column in Table S1 in Supplementary file 1.

– Supplementary Figures 1 and 2: A "partial" (Tulán 54) and "panoramic" (Tulán 84) site views are provided. However, it would be useful to include a plan of both sites to know the spatial provenience of the bones discussed at a more detailed scale.

The information of each sample was included in Table S1 in Supplementary file 1. We consider that it is not relevant to include a spatial plan of each site because our main goal was more focused on DNA analyses of camelid bones.

– In the subsection “Biogeographical area”, different dates are provided for the Formative period; it would be clearer to state that it starts at c. 5000 BP and the period represented in these particular sites is c. 3360-2370 BP. The inference that human populations started selecting wild camelids to raise their own herds c. 3000 years ago should be supported with some references.

All ^14^C ages have now been included in the main text and references included to support previous works. Additional references were included in the text:

Cartajena, 2013; Cartajena et al., 2019; Izeta, 2004; Loyola, Núñez and Cartajena, 2019a; Núñez, Grosjean and Cartajena, 2002; Núñez et al., 2005; Núñez et al., 2017a.

3) Explicit discussion of limitations: The authors should be certain to clearly state the caveats of this study. They should be clear about the potential influence of factors, and their timing, that are independent of domestication on both size and molecular diversity. Specifically, these are limitations imposed by:– The osteometric and aDNA methods.

The limitations of ancient DNA analyses were added in the Discussion, subsection “Early Hybridization of domesticated camelids”.

The discussion about osteometric methods and their validity for taxonomic identification is explained in the Results.

– The archaeological sample relative to the research question (e.g. the chronological and geographical representation of the selected sample).

We included an extended description of the sites Tulán-54, Tulán-85, Tulán-52 and Tulán-94, subsection “Archaeological samples”. We explained why we chose samples from that period of time, Early Formative and why the Tulán ravine in San Pedro de Atacama is a suitable place for looking at the domestication process of SAC from early times. Materials and methods.

Moreover, we have included pictures of the sites and a map with the location of the sites. Figure 1—figure supplement 1.

– Hybridization and later killing-off events.

The problem with the hybridization and later killing-off events is explained in the Discussion, section “Early Hybridization of domesticated camelids.”

– Potential impact of climatic/environmental changes.

We refer to the impact of the climatic changes in the Discussion.

– The extent to which any of the above is knowable.

The exact extent as to which any of the above mentioned limitations is knowable is somewhat limited by our dataset as it is just a single marker from a relatively small geographical region. While we have still added some significant new hypotheses to the question of SAC domestication and evolution, the most comprehensive way to address these would be through the inclusion of a large palaeogenomic dataset from multiple sites across the Andes. We mention that the inclusion of ancient nuclear genomes may shed some light on these with more depth, however, such data is out of the scope of the current study. Discussion.

4) Interpretations of hybridization:In the subsection “Early Hybridization of domesticated camelids”, if modern genomes are extensively hybridized in the recent past, how valid are the modern samples used in this analysis? In other words how can a molecular taxonomic identification actually work – and be compared against the morphological results – if the modern reference collection of, say, a "guanaco" might actually have been hybridized in the recent past? And if only mitogenomes are used in this analysis, how can such hybridization truly be pulled apart? Then, the specific "hybrid specimens" described here are not really directly addressed in the Results sections; on what basis are these individuals determined to be hybrids? If it is simply the difference between morphology and mitogenome haplotype (which seems to be the case), then we should investigate which specimens were first phalanges and which ones were astragali, because the astragali seem to be much more effective at separating them out (see point #1). On the trees that code the specimens by size, I am reading this to mean that anything in the "intermediate" category was a phalanx, and therefore I'm not sure that this is a "hybrid" so much as a problem with the discrimination of the method. Alternatively, if they are truly hybrids, then there might not be expected to be a clear morphological result from the bone measurements.

The hybridisation history of South American Camelids is focused around the hybridisation events that occurred between alpacas and llamas, i.e. between the domestic forms and not between the wild camelids (Guanacos and Vicuñas). Hence any statement reflecting hybridisation events in the past are limited to those that occurred between llamas and alpacas.

Three main sections for this argument:

1) We do find significant differences between large and small animals with morphology. Discussion.

The bones from which no DNA was extracted were removed from the analyses as their results bear little consequence to our work presented here (we updated Figure 2 accordingly). Additionally, we carried out statistical tests between the different measurements carried out for each of the bones and showed that the large specimens are significantly different from the small specimens (all p-values were smaller than 10^-4^).

2) Hybridization is largely a new thing and not an old event.

“In particular, Fan et al., 2020, timed the hybridization history between South American camelids and found evidence for a post-colonial hybridization, and not an ancient one, thereby suggesting that the bulk of the taxonomic uncertainty driven by this process affects modern samples. Moreover, although rare, hybridization between llamas and guanacos (i.e. large bodied animals), and separately between alpacas and vicuñas (small bodied animals) have been previously reported (González et al., 2020). While the results presented here do not disagree with the hybridization patterns described elsewhere (i.e. the recent hybridization history timed by Fan et al., 2020), the observation of hybrid animals among the zooarcheological samples (i.e. those samples that present a discordance between their taxon identification based on bone morphology and their molecular taxonomic identification), suggests that the hybridization history of the domestic camelids is yet far from understood, and could have also occurred prior to the European colonization of South America.”

3) It is possible that Clades 1-3 are mostly guanacos and not domestic animals:

“Furthermore, although the wild, a hybrid lineages can consistently may carry the same mitochondrial haplotype throughout time, they may change body size (e.g. become larger) through backcrossing as the host population genome becomes prevalent in the lineage due to mendelian inheritance. […] However, the presence of both domestic camelids in Tulán has been previously established through the analyses of bone remains (Cartajena et al., 2007; Núñez et al., 2017), microscopic camelid fiber analyses (Benavente, 2005, 2006), and the analyses of bone collagen stable isotopes that differentiate llamas ritually fed with maize from wild guanacos (López, Cartajena and Núñez, 2013).”

5) Discussion of implications of bottlenecks:– It is interesting that the vicuña, which according to the model reconstructed here was not a major source of domesticated stock in the past, is the one that has undergone the most genetic bottlenecking; could this potentially be as a result of human manipulation of other taxa that caused their populations to increase at the expense of wild vicuña habitat? What is the actual mechanism that the authors propose for the bottleneck in vicuña?

We hypothesize that the stronger loss of variation in the vicuña is the outcome of the preferential hunting of vicuñas rather than guanacos by the early upper Andean communities, reflecting aspects of the ecology and behavior of vicunas:

“This is not surprising when considering the relative homogeneity of the environment where vicuñas are found and how gregarious they are in comparison to guanacos. […] Such aridification is likely to have changed the habitable range for the vicuña further exacerbating the demographic pressure that it was already experiencing.”

– The Holocene SAC demographic decline is related to human behavior in the manuscript. It would be interesting to assess the impact of climatic and concomitant environmental changes since the late Pleistocene – including the Antarctic Cold Reversal and the Hypsithermal – on their demography as well. The reasons for vicuñas undergoing bottlenecks unlike guanacos is also worth further exploring. It will also be interesting to explore why vicuña demography has recovered so nicely in recent years in spite of that, while Andean and Chaco guanacos – excluding Fuego-Patagonian steppe populations – are now so patchily distributed that are considered to be endangered. It would be interesting to know how genetic variation corresponds to camelid subspecies in modern samples, and whether this variation can be traced back in the fossil samples.

We thank the reviewers for highlighting the Antarctic Cold Reversal and Hypsithermal, which we have now addressed in our paper (see point above).

The issue of the recovery of the vicuña and the endangerment of the guanaco was also addressed in the last paragraph of the Discussion.

The relationship between haplotypes shared by the ancient and modern samples is shown in Table S9 in Supplementary file 1.

6) Adjustments to figures:– Figure 1: The map legend should stress on what kind of data it is based, whether on the documented provenience localities where the subspecies have been previously described from a taxonomic point of view, or from any other source of information. It would be also helpful to have some precisions on the time period represented: pre or postcolonial times. The sources of the distributional information should be cited. More importantly, the fact that there are these subspecies and their main differences should be introduced when presenting SACs and before the figure. The color-coding used for the map is not enough clear to distinguish the overlapping distribution of these four subspecies, as is the case of the southern vicugna and the guanaco. Also, there are extant guanaco populations in Córdoba, Santiago del Estero and Buenos Aires provinces in Argentina that do not appear in Figure 1. The map should show the georeferenced points from where the specimens used in the analysis (data from Table S3 in Supplementary file 1). The apparent attribution to the L. g. guanicoe subspecies in the map is a supposition since the guanaco of Northwest Argentina has not been studied genetically yet. As L.g. cacsilensis is present in southern latitudes (for example, at Ovalle and Paposo in Chile, according to Table S3 in Supplementary file 1), distant from its northern type locality, and the Andes appear "to offer little impediment to movement" (González et al., 2006: 170) it would be important to establish if in the past there has been an interconnection between both guanaco populations.

The Figure 1 was completely replaced by a new one with a more precise distribution of the current distribution of the guanaco (A), separated from the current distribution of the vicuña (B). The colors used in this figure are in correspondence with the rest of the figures along the manuscript.

This figure shows the collection of modern samples from the places where the mitogenomes and the Control Region were obtained. Regarding the limit of the guanaco subspecies, there is a very wide contact zone that includes the northeast of the distribution, Monte Argentino and North-central Chile, but whose individuals belong predominantly to L. *g. guanicoe* (Marín et al., 2013) so we assume that the guanacos of Northeastern Argentina are as well.

– Figure 2: In the section "Morphometric analysis" it says that measurements were taken from 61 specimens out of a sample of 75 bones, and that of those measured specimens 18 were not considered for morphometric analysis. In a previous section it says that the mitogenomes dataset comprises 60 ancient records. These numbers do not add up. This seems to suggest that DNA was extracted from bones that were not included in the morphometric analysis. If this is true it should be mentioned. It would give a better understanding of the complementary nature of the osteometric and genetic database and the research design. The dots are also difficult to differentiate, because the labels overlap so much and the dots are the same size and color; which ones are archaeological and included versus not included in the analysis? There also seem to be no modern camelid measurements for the astragalus?

We fixed the mistake of the dataset, there are a total of 61 ancient mitogenomes, review Table S5 in Supplementary file 1.

Figure 2A and B were remade and removed the dots that were not included in the ancient DNA analysis. Moreover we included modern camelid measurements for the astragalus for the four species in Figure 2B.

– Figure 3: Even with a zoom 200% it is impossible to read the posterior probability values of the support of the nodes. It is interesting that there are a number of "small-sized" archaeological specimens in Clade 2. Are these samples on astragali (better indicators) or phalanges (more variation)? There is a lot more variation in the sizes of adult phalanges and astragali in the archaeological sample than in the modern sample, and potentially this may have something to do with active moments of admixture and hybridization between the different clades. This relationship between the morphological data from the two different elements and how this would be represented under various hybridization scenarios could be better-addressed in the text (see point #4).

We are sorry for having submitted the low font version of the figure. Because the majority of the posterior probability values for the nodes’ supports are larger than 0.9, we opted for removing all support values of the three and only indicate those with a posterior probability support lower than 0.9. We have accordingly added a line to the figure legend indicating this and the percentage of the nodes that have a posterior probability larger than 0.9.

We have added information on the sample sizes used to estimate the standard deviations obtained from bone measurements in the modern samples, which for three of the species correspond to 7 or fewer individuals, and only the llama had 14 samples measured. The issue about hybridisation scenarios was addressed in point 4.

– Figure 5 is really interesting and easy to read, but surprising that the small-sized archaeological samples group exclusively away from the vicuña.

For this network we only used the most variable region of the mitochondrial DNA. The results could have changed compared with the complete mitochondrial genome phylogeny.

– Figure 6: Is there some way to indicate which taxa are associated with which haplogroups here, as well as linking the past with the present? It is a really useful figure, but difficult to understand the relationships between taxa (even if only the modern taxa are indicated). Perhaps the (lost) vicuña haplotypes be indicated by A vicuña figure, similar to Figure 5 – which is very clear.

We added Table S9 in Supplementary file 1, with the information about which taxa are associated with which haplogroups and the ancient samples included in each haplotype.

– Figure 7: the labelled panels (A, B, etc.) do not have any specific information about the samples to which they pertain.

For this figure we used random modern samples conditional on the sampling size of the ancient DNA samples. The purpose of this analysis was to measure the genetic diversity between modern and ancient samples.

– Supplementary Figure 3: A minor thing maybe, but the underlying grey map seems to have different square shades of grey. This is a bit irritating. The legend next to the map is difficult to read, the quality of the letters is not good and the green and blue dots have grey squares around them.

Supplementary Figure 3 was removed. Nevertheless, location of the archaeological site and map of modern camelid samples included in the Bayesian phylogenetic analysis was included in the new Figure 1.

7) References:Check for missing references, incomplete ones and errors in them. There are some inconsistencies in how the authors are listed in the text and in the Bibliography, particularly with authors who have two last names/surnames. For example: "López, P." and "López Mendoza, P." are the same person; while "Mengoni-Goñalons" correctly in the text, is only "Goñalons" in the Bibliography. The authors might also consider how their names are listed here with respect to how they have published their names in other papers.

We have checked and fixed the errata in the bibliography.

8) Formatting:– Although it is possible to place Materials and methods to be near the front of the manuscript, eLife style does typically place it at the end.

Materials and methods has been moved to the end of the manuscript before the bibliography.

– The supplementary information should also appear together at the end of the manuscript, not divided up into sections within the main body of the manuscript.

Supplementary material has been moved to the end of the manuscript.

– Check for typos and formatting errors; date ranges should always be provided from the older to the younger age. Also check for grammar and sentence structure across the text, as many phrases need rewriting (for instance, that in the Abstract reading "we generated mitochondrial genomes for… 815 extant mitochondrial control region sequences from across South America"; "Northeastern Argentina" should read "Northwestern Argentina," etc.). The text needs extensive revision in this regard.

Thank you for the observation. We have checked and reviewed those errata.

– Figures and tables are not numbered after the order in which they appear in the text, and formatting is not even (e.g., way to name pictures within one figure). While the text mentions that first phalanges were measured, the figures and tables should also read so (and not just "phalanx"). Letters in the pictures making up Supplementary Figure 2 do not match the legend; they mix with Supplementary Figure 1. Figure 3—figure supplement 3 is only mentioned in the legend of another figure and not in the main text.

We have checked and reviewed those errata.